# MyD88 promotes myoblast fusion in a cell-autonomous manner

Sajedah M. Hindi[1], Jonghyun Shin[1], Yann S. Gallot [1], Alex R. Straughn[1], Adriana Simionescu-Bankston[1], Lubna Hindi[1], Guangyan Xiong[1], Robert P. Friedland[2] & Ashok Kumar[1]

Myoblast fusion is an indispensable step for skeletal muscle development, postnatal growth, and regeneration. Myeloid differentiation primary response gene 88 (MyD88) is an adaptor protein that mediates Toll-like receptors and interleukin-1 receptor signaling. Here we report a cell-autonomous role of MyD88 in the regulation of myoblast fusion. MyD88 protein levels are increased during in vitro myogenesis and in conditions that promote skeletal muscle growth in vivo. Deletion of MyD88 impairs fusion of myoblasts without affecting their survival, proliferation, or differentiation. MyD88 regulates non-canonical NF-κB and canonical Wnt signaling during myogenesis and promotes skeletal muscle growth and overload-induced myofiber hypertrophy in mice. Ablation of MyD88 reduces myofiber size during muscle regeneration, whereas its overexpression promotes fusion of exogenous myoblasts to injured myofibers. Our study shows that MyD88 modulates myoblast fusion and suggests that augmenting its levels may be a therapeutic approach to improve skeletal muscle formation in degenerative muscle disorders.

[1] Department of Anatomical Sciences and Neurobiology, University of Louisville School of Medicine, Louisville, KY 40202, USA. [2] Department of Neurology, University of Louisville School of Medicine, Louisville, KY 40202, USA. Sajedah M. Hindi, Jonghyun Shin, and Yann S. Gallot contributed equally to this work. Correspondence and requests for materials should be addressed to A.K. (email: ashok.kumar@louisville.edu)

Skeletal muscle is composed of bundles of contractile myofibers. Each myofiber is a syncytium that arises by the fusion of hundreds or thousands of mononucleated myoblasts during embryonic development of skeletal muscle[1, 2]. Myoblast fusion is a critical step not only for development of skeletal muscle during embryogenesis, but also for satellite cell-mediated regeneration of injured myofibers in adults[3, 4]. Moreover, myoblast fusion precedes an increase in skeletal muscle size during overload-induced muscle hypertrophy[5–8].

Myoblast fusion involves migration of fusion-competent myogenic cells, alignment of their membranes, rearrangements of the cytoskeleton at contact sites, opening of fusion pores for exchange of cytoplasmic material, and ultimately merging of two myogenic cells into one[3, 4, 9–13]. While it was initially considered that myoblast fusion is tightly coordinated with the muscle differentiation program and requires the expression of myogenic regulatory factors (MRFs) in at least one fusion partner[3, 11, 14],

recent studies have provided evidence that fusion can also proceed in the absence of MRFs. Three independent studies have recently identified a muscle-specific 84-amino acid peptide, named Myomixer, which is essential for myoblast fusion in vitro and skeletal muscle formation during embryogenesis[15–17]. Importantly, Myomixer along with Myomaker can also induce fibroblast–fibroblast fusion suggesting that this two component muscle-specific system can bestow fusogenic activity to non-myogenic mammalian cells[15–17].

Myoblast fusion involves an array of signaling pathways that are activated as a result of the recruitment of specific cell-surface proteins between fusion partners or as a part of the myogenic differentiation program[14, 18]. Wnt signaling is essential for muscle development during embryogenesis, postnatal myogenesis, and myoblast fusion[19]. Activation of the canonical Wnt pathway promotes myoblast fusion in vitro[20] and during regeneration of skeletal muscle in adult mice[21]. The nuclear factor kappa-B (NF-

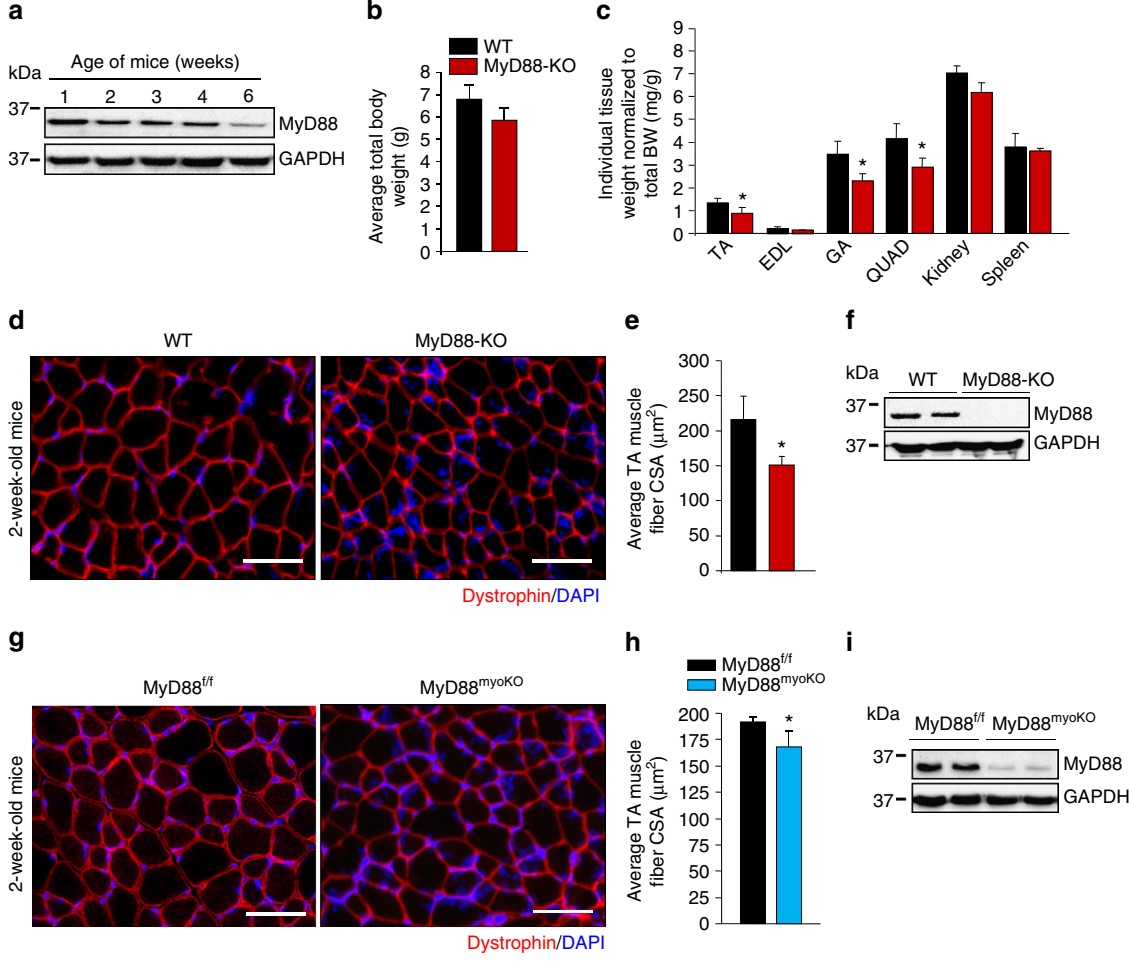

**Fig. 1** MyD88 regulates postnatal growth of skeletal muscle. **a** Gastrocnemius (GA) muscle was collected from WT mice at indicated ages and protein extracts made were immunoblotted for MyD88 and an unrelated protein GAPDH. **b** Total body weight of 2-week-old WT and MyD88-KO mice. **c** Tibialis anterior (TA), extensor digitorum longus (EDL), gastrocnemius (GA), and quadriceps (QUAD) muscles, kidney, and spleen were isolated from 2-week-old WT and MyD88-KO mice and their individual wet weight normalized to body weight is presented here. $N = 4$ mice in each group. *$p < 0.05$ from WT mice by unpaired $t$-test. **d** TA muscle from 2-week-old WT and MyD88-KO animals was processed for histological and morphometric analysis. Representative photomicrographs of dystrophin-stained TA muscle sections from WT and MyD88-KO mice. Scale bar: 50 μm. **e** Quantification of average myofiber cross-sectional area (CSA) in TA muscle of 2-week-old WT and MyD88-KO mice. $N = 4$ mice in each group. *$p < 0.05$ from WT mice by unpaired $t$-test. **f** Immunoblot presented here demonstrates absence of MyD88 protein in skeletal muscle of MyD88-KO mice. **g** Representative photomicrographs of dystrophin-stained TA muscle sections from 2-week-old MyD88$^{f/f}$ and MyD88$^{myoKO}$ mice. Scale bar: 50 μm. **h** Quantification of average myofiber CSA in TA muscle of 2-week-old MyD88$^{f/f}$ and MyD88$^{myoKO}$ mice. $N = 4$ in each in each group. Error bars represent s.d. *$p < 0.05$ from MyD88$^{f/f}$ mice by unpaired $t$-test. **i** Levels of MyD88 and unrelated protein GAPDH in skeletal muscle of MyD88$^{f/f}$ and MyD88$^{myoKO}$ mice measured by performing western blot

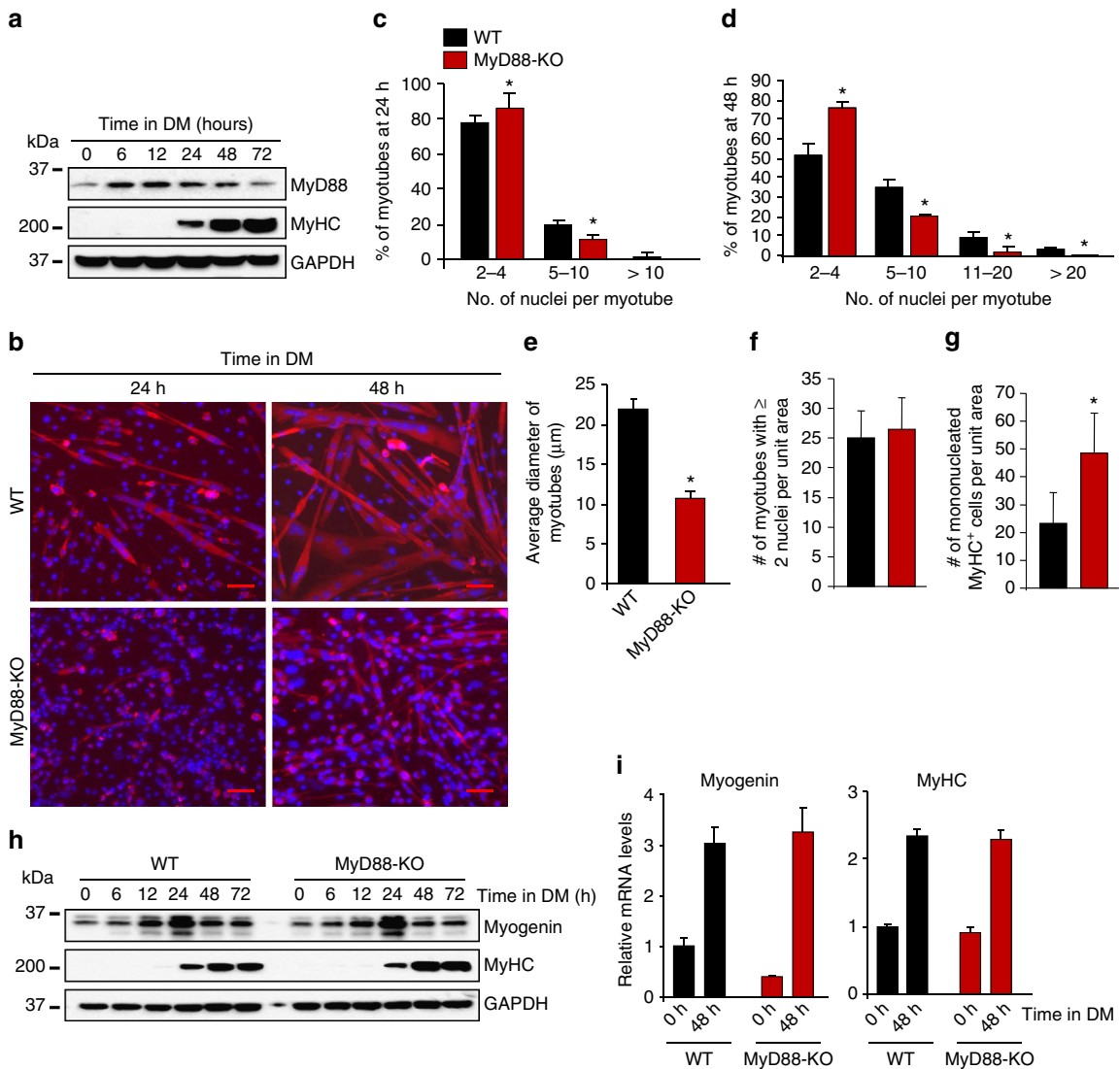

**Fig. 2** MyD88 mediates myoblast fusion in vitro. **a** Primary myoblasts were prepared from hind limb muscles of WT mice. The cells were incubated in DM and samples were collected at indicated time points and processed by Western blot to detect MyD88, MyHC as a differentiation marker, and an unrelated protein GAPDH. **b** Primary myoblasts prepared from hind limb muscle of WT and MyD88-KO mice were plated at equal densities and incubated in DM for 24 h or 48 h followed by staining for MyHC and DAPI. Representative photomicrographs are presented here. Scale bar: 50 μm. **c** Quantification of the percentage of myotubes containing indicated number of nuclei per myotube in WT and MyD88-KO myotubes cultures after 24 h of addition of DM. **d** Quantification of the percentage of myotubes containing indicated number of nuclei per myotube in WT and MyD88-KO cultures after 48 h of addition of DM. **e** Average diameter of myotubes in WT and MyD88-KO cultures after 48 h of addition of DM. **f** Quantification of number of MyHC+ myotubes containing 2 or more nuclei and **g** number of mononucleated MyHC+ cells per unit area (~0.15 mm²) in WT and MyD88-KO cultures after 48 h of incubation in DM. **h** Primary myoblasts prepared from WT and MyD88-KO mice were incubated in DM and samples were collected at indicated time points. Protein lysates were prepared and blotted with antibodies against MyHC, myogenin and unrelated protein GAPDH. **i** In a separate experiment, WT and MyD88-KO cells were collected at 0 and 48 h following addition of DM and mRNA levels of MyHC and myogenin were measured after normalizing to the levels of β-actin. Results are from four to five independent experiments. Error bars represent s.d. *$p < 0.05$ from corresponding WT cultures for all experiments by unpaired $t$-test

κB) transcription factor is an important regulator of myogenesis[22]. Depending on the stimulus, NF-κB can be activated through the canonical or non-canonical pathway[23]. The canonical NF-κB pathway is constitutively activated in proliferating myoblasts but repressed during differentiation of myoblasts into multinucleated myotubes. By contrast, the non-canonical NF-κB pathway becomes activated during myogenic differentiation[24, 25]. It was recently reported that the non-canonical NF-κB signaling promotes myoblast fusion during myogenesis[26]. In addition, a few other signaling cascades such as MEK5/ERK5, Integrin/focal adhesion kinase (FAK), Rho family GTPases, and calcineurin

nuclear factor of activated T cells 2 (NFATc2) promote myoblast fusion during myogenesis[27–34]. However, apical signaling mechanisms that regulate the activation of various profusion pathways during myogenesis remain unknown.

The MyD88 is the key adaptor protein for the interleukin-1 receptor (IL-1R) and most Toll-like receptors (TLRs), which are essential for innate immunity and pathogen-associated molecular pattern recognition[35–37]. Upon recruitment to the TLRs, MyD88 forms a complex with IL-1R-associated kinase (IRAK) 4, IRAK1, and IRAK2, leading to the activation of IRAK4, which subsequently phosphorylates IRAK1 and IRAK2. Phosphorylated

IRAK1/2 then mediates the activation of TNF-receptor associated factor 6 (TRAF6), an E3 ubiquitin ligase, which conjugates K63-linked ubiquitin chains to itself and other proteins in cooperation with the E2 dimer Ubc13/Uev1A[38]. TRAF6 ubiquitylates and activates TGF-β activated kinase 1 (TAK1), which in turn activates inhibitor of kappa-B (IκB) kinase (IKK), JNK, and p38 MAPKs, ultimately leading to the activation of transcription factors, NF-κB and activator protein 1, and induction of inflammatory molecules[36, 37]. Mutations in the MyD88 gene lead to the development of cancer in humans and mice suggesting that MyD88 also plays a cell autonomous role in tissue homeostasis[39, 40]. However, the role of MyD88 in the regulation of myogenesis has not yet been investigated.

In the present study, we demonstrate that MyD88 regulates the myoblast fusion step of myogenesis through stimulating non-canonical NF-κB and canonical Wnt signaling pathways. Ablation of MyD88 inhibits muscle growth during postnatal development, overload-induced myofiber hypertrophy, and size of regenerating myofibers in mice. Conversely, overexpression of MyD88 in exogenous myoblasts improves their fusion to injured myofibers. Overall, our study demonstrates that enhancing MyD88 levels in myoblasts may be a potential approach to improve skeletal muscle formation in diverse conditions.

## Results

**MyD88 promotes skeletal muscle growth post-birth.** Our initial analysis showed that MyD88 protein was present in high amounts in skeletal muscle of neonatal or young wild-type (WT) mice and was gradually reduced during development to the adult stage (Fig. 1a). Although not statistically significant, body weight of 2-week-old MyD88-KO mice was reduced compared to age-matched WT mice (Fig. 1b). Interestingly, wet weight of the individual tibialis anterior (TA), gastrocnemius (GA), and quadriceps (QUAD) muscle, but not kidney and spleen, was significantly reduced in 2-week-old MyD88-KO mice compared with age-matched WT mice (Fig. 1c). We next generated transverse sections of TA muscle from 2-week-old WT and MyD88-KO mice and immunostained them for dystrophin protein. Consistent with wet weight, the circumference of TA muscle of MyD88-KO mice appeared smaller compared to WT mice (Supplementary Fig. 1A). Quantitative analysis of dystrophin-stained sections showed that the average myofiber cross-sectional area (CSA) was significantly reduced in TA muscle of 2-week-old MyD88-KO mice compared with age-matched WT mice (Fig. 1d–f).

We also generated myoblast-specific MyD88-KO mice (henceforth MyD88[myoKO]) by crossing floxed MyD88 (MyD88[f/f]) with Myod1-Cre mice. Similar to whole-body MyD88-KO mice, average myofiber CSA was significantly reduced in TA muscle of 2-week-old MyD88[myoKO] mice compared to littermate MyD88[f/f] mice (Fig. 1g–i). To understand whether MyD88 regulates the skeletal muscle formation, we first performed immunostaining of TA muscle sections for embryonic isoform of myosin heavy chain (eMyHC) or MyoD protein. However, there were no eMyHC[+] myofibers or MyoD[+] nuclei in TA muscle of 2-week-old MyD88[f/f] or MyD88[myoKO] mice. Immunostaining for Pax7 protein (a marker of satellite cells) showed that there was no significant difference in number of Pax7[+] cells in TA muscle of 2-week-old MyD88[f/f] and MyD88[myoKO] mice (Supplementary Fig. 1B, 1C).

We next generated TA muscle sections from 5-day-old (P5) MyD88[f/f] and MyD88[myoKO] mice and immunostained them for eMyHC and laminin protein. Nuclei were identified by staining with 4′,6-diamidino-2-phenylindole (DAPI). Interestingly, we found that there was an increased number of smaller size mononucleated eMyHC[+] myocytes in transverse sections of TA muscle of 5-day-old MyD88[myoKO] mice compared with MyD88[f/f] mice (Supplementary Fig. 2A, 2B). To specifically investigate whether MyD88 regulates myoblast fusion during skeletal muscle growth post-birth, we performed an 5-Ethynyl-2′-deoxyuridine (EdU) incorporation assay. At day 5 post-birth (P5) and P8, MyD88[f/f] and MyD88[myoKO] mice were given intraperitoneal injections of EdU. The mice were euthanized at the age of 2 weeks and TA muscle was isolated and processed for the detection of EdU[+] nuclei within myofibers. Interestingly, the number of intramyofiber EdU[+] nuclei was significantly reduced in 2-week-old MyD88[myoKO] mice compared to littermate MyD88[f/f] mice (Supplementary Fig. 3A, 3B). These results indicate that MyD88 may mediate myoblast fusion during postnatal growth.

Although we observed a reduced myofiber size in 2-week-old MyD88-KO and MyD88[myoKO] mice compared to their corresponding control mice, there was no significant difference in myofiber CSA at the age of 3 months. This could be attributed to multiple reasons, including the possibility that myonuclear accretion proceeds at a slower rate in MyD88[myoKO] mice, but ultimately there is a comparable number of terminally fused myonuclei in myofibers of MyD88[f/f] and MyD88[myoKO] mice. To investigate such possibility, we generated single myofiber cultures from skeletal muscle of 3-month-old MyD88[f/f] and MyD88[myoKO] mice and quantified the number of nuclei after staining with DAPI. Interestingly, the number of nuclei per unit length was significantly reduced in myofibers of 3-month-old MyD88[myoKO] mice compared with littermate MyD88[f/f] mice (Supplementary Fig. 4A, 4B). These results suggest that even though MyD88[myoKO] myofibers contain a significantly reduced number of myonuclei compared to MyD88[f/f], this number is sufficient to support growth resulting in comparable myofiber CSA in skeletal muscle of 3-month-old MyD88[f/f] and MyD88[myoKO] mice.

**MyD88 promotes fusion of cultured myoblasts during myogenesis.** We first investigated how the levels of MyD88 are regulated during myogenesis in vitro. Interestingly, the levels of MyD88 protein were considerably increased starting at 6 h of incubation of primary myoblasts in differentiation medium (DM) and remained elevated up to 48 h (Fig. 2a). There was no significant change in mRNA levels of MyD88 at different time points after addition of DM (Supplementary Fig. 5A), suggesting that during myogenesis MyD88 is upregulated through post-transcriptional mechanisms that are yet to be identified. To understand the role of MyD88 in the regulation of various steps of myogenesis, we prepared primary myoblasts from WT and MyD88-KO mice and studied their proliferation, differentiation, and fusion. There was no significant difference in the proliferation of WT and MyD88-KO myoblasts, measured by EdU incorporation assay (Supplementary Fig. 5B, 5C). Our experiments also showed that there was no significant difference in the levels of lactate dehydrogenase (a marker of cell mortality[38, 41]) in culture supernatants of WT and MyD88-KO myoblasts at 48 h after incubation in DM suggesting that deletion of MyD88 does not affect the survival of myogenic cells (Supplementary Fig. 5D).

We next investigated whether MyD88 regulates differentiation of myoblasts into myotubes. Primary myoblasts prepared from hind limb muscles of WT and MyD88-KO mice were incubated in DM for 24 h or 48 h and the cultures were immunostained for MyHC. Nuclei were counterstained with DAPI. As shown in Fig. 2b, myotube formation was drastically inhibited in MyD88-KO cultures compared to WT cultures. Quantification of number of nuclei per myotube showed that compared to WT cultures, MyD88-KO cultures had significantly higher number of myotubes containing 2–4 nuclei, but had a significantly reduced

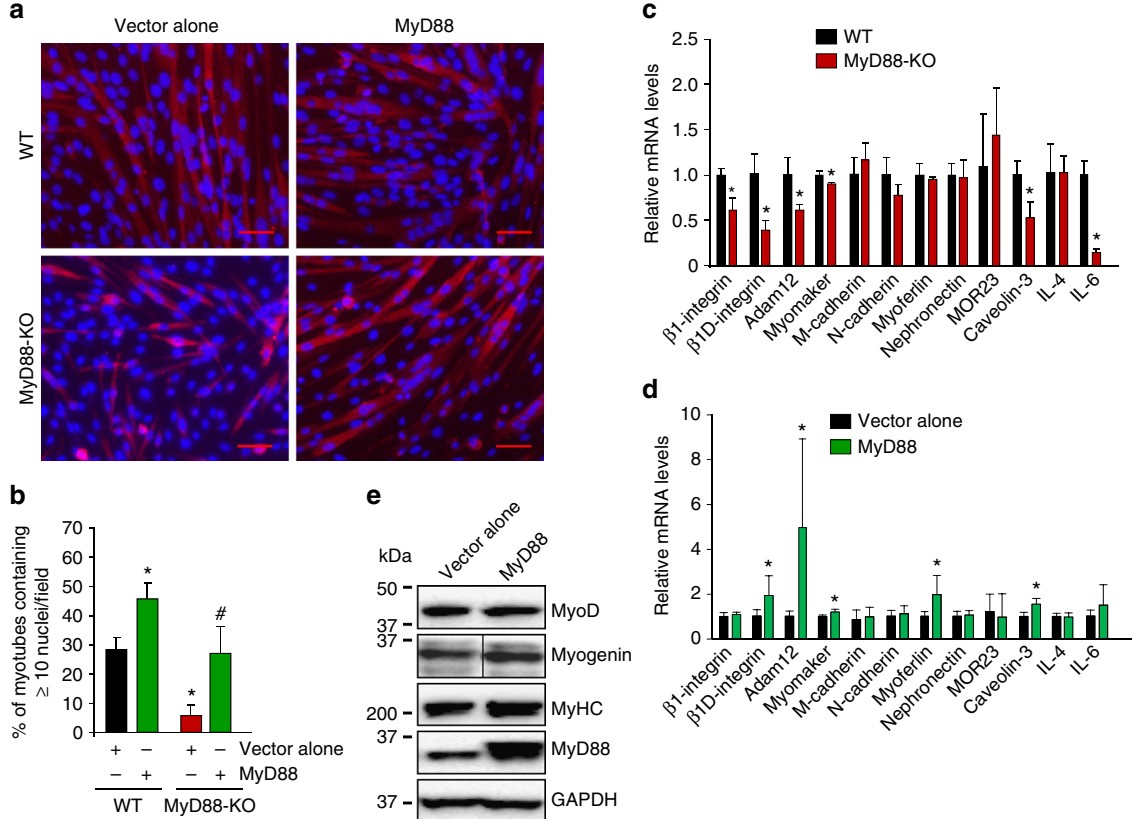

**Fig. 3** MyD88 promotes myoblast fusion through augmenting the expression of profusion molecules. **a** Primary myoblasts prepared from hind limb muscle of WT and MyD88-KO mice were transfected with empty vector or vector containing MyD88 cDNA. The cells were incubated in DM for 48 h followed by staining with anti-MyHC and DAPI. Scale bar: 50 μm. **b** Quantification of the percentage of myotubes containing ≥ 10 nuclei in WT and MyD88-KO cultures. $N = 5$ in each group. Error bars represent s.d. *$p < 0.01$ from WT cultures transfected with vector alone by unpaired $t$-test. #$p < 0.01$ from MyD88 cultures transfected with vector alone by unpaired $t$-test. **c** Primary myoblasts prepared from hind limb muscle of WT and MyD88-KO mice were incubated in DM for 48 h after which samples were processed for QRT–PCR analysis to measure relative mRNA levels of indicated profusion molecules. $N = 3$ in each group. *$p < 0.05$ from WT cultures by unpaired $t$-test. Primary myoblasts were prepared from hind limb muscle of WT mice and transfected with empty vector or vector containing MyD88 cDNA. Cells were then incubated in DM for 48 h and **d** processed for gene expression of indicated profusion molecules using QRT–PCR assay. $N = 5$–6, *$p < 0.05$ from cultures transfected with vector alone by unpaired $t$-test. **e** Protein lysates were prepared and blotted for MyoD, myogenin, MyHC, MyD88 and unrelated protein GAPDH. Vertical black line indicates that intervening lanes were spliced out

number of myotubes containing 5 or more nuclei (Fig. 2c, d). Furthermore, the average myotube diameter was significantly reduced in MyD88-KO cultures compared to WT cultures (Fig. 2e).

During myogenesis, fusion occurs in two phases. Initially, fusion-competent mononucleated myoblasts fuse with each other (i.e. primary fusion) resulting in the formation of nascent myotubes. In the second phase, myoblasts fuse with nascent myotubes (i.e. secondary fusion) resulting in nuclear accretion and growth of myotubes[3, 4, 14]. We found that there was no significant difference in the number of myotubes containing two or more nuclei. However, there was a significant increase in the number of mononucleated MyHC+ cells in MyD88-KO cultures compared to WT cultures at 48 h after addition DM (Fig. 2f, g) suggesting that MyD88 primarily mediates secondary myoblast fusion during myogenesis.

We also evaluated whether MyD88 regulates the expression of differentiation markers during myogenesis. Interestingly, there was no significant difference in the protein levels of myogenin or MyHC between WT and MyD88-KO cultures at different time points after the addition of DM (Fig. 2h). Furthermore, mRNA levels of myogenin and MyHC were comparable between WT and MyD88-KO cultures at 48 h after addition of DM (Fig. 2i)

suggesting that MyD88 does not affect differentiation of myoblasts into myotubes.

By performing a cell mixing experiment, we next investigated whether MyD88 mediates myoblast fusion when present in one or both fusion partners. Results showed that while MyD88-KO myoblasts (expressing mCherry) fuse with WT myoblasts (expressing green fluorescent protein, GFP), the fusion efficacy was markedly reduced resulting in the reduced number and smaller size chimeric myotubes (Supplementary Fig. 6A, 6B).

To confirm the role of MyD88 in myoblast fusion, we also employed a siRNA approach. Primary myoblasts prepared from WT mice were transfected with scrambled (control) siRNA or two distinct siRNAs targeting two different regions of the MyD88 mRNA. Results showed that both MyD88 siRNAs dramatically reduced the formation of multinucleated myotubes in primary myoblast cultures (Supplementary Fig. 7). Collectively, these results suggest that MyD88 promotes myoblast fusion during myogenesis.

**Overexpression of MyD88 augments myoblast fusion**. We next investigated the effects of overexpression of MyD88 on myoblast fusion. WT or MyD88-KO myoblasts were transfected with empty vector or vector containing MyD88 cDNA. The cells were then incubated in DM for 48 h and immunostained for MyHC.

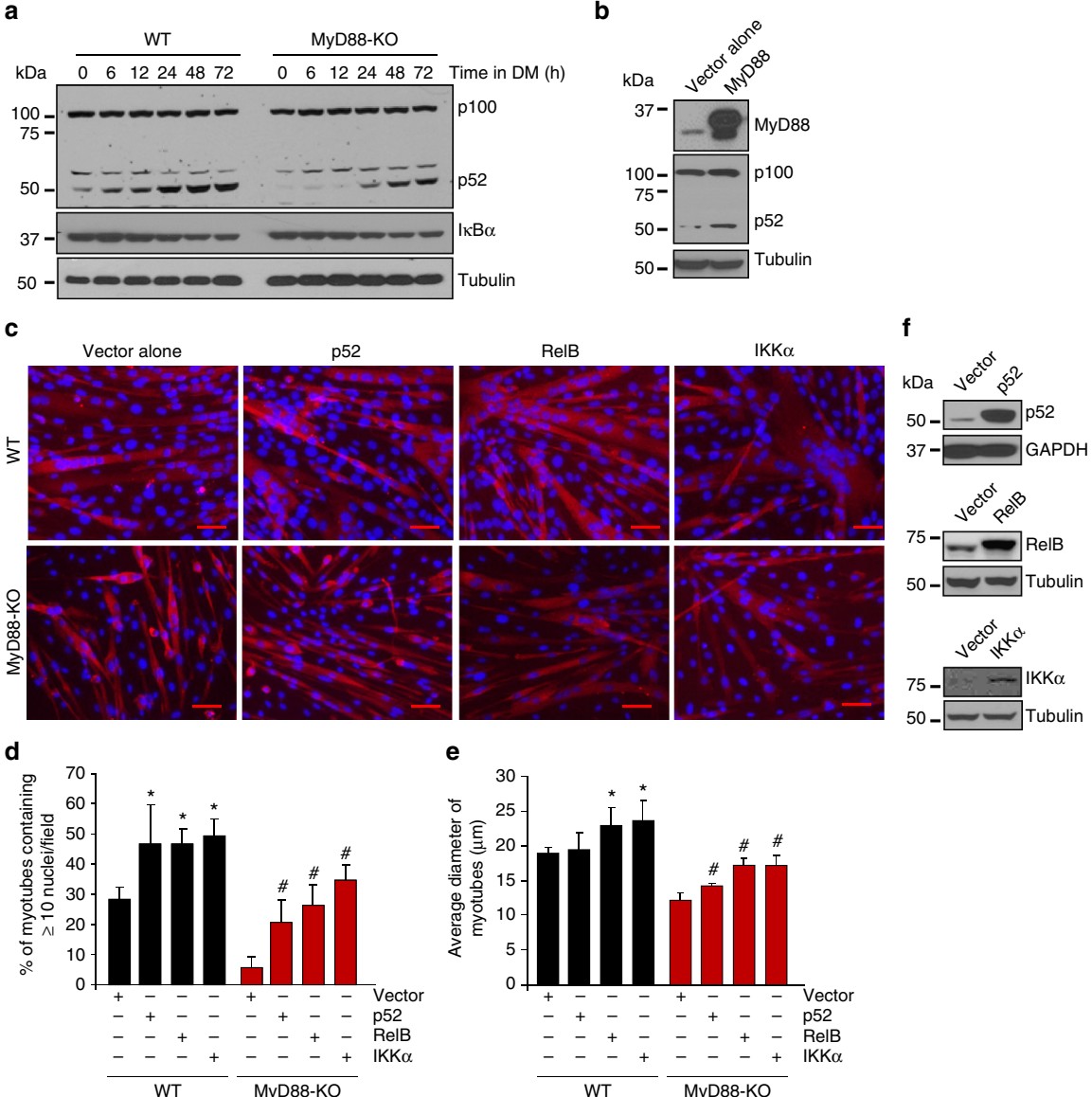

**Fig. 4** MyD88 regulates myoblast fusion through non-canonical NF-κB pathway. **a** Primary myoblasts were prepared from hind limb muscle of WT and MyD88-KO mice. Cells were then plated at equal densities and incubated in DM. Samples were collected at various time points following addition of DM and processed for biochemical analysis. Immunoblots demonstrate the levels of p100/p52, IκBα, and unrelated protein tubulin. **b** Primary myoblasts prepared from WT mice were transfected with empty vector or vector containing MyD88 cDNA. After 48 h, cells were collected and protein lysates were made. Representative immunoblots demonstrating levels of MyD88, p100/p52, and unrelated protein tubulin. **c** WT and MyD88-KO primary myoblasts were transfected with vector alone or vector containing cDNA for p52, RelB, or IKKα. The cells were incubated in DM for 48 h and cultures were fixed and stained with anti-MyHC and DAPI. Representative photomicrographs are presented here. Scale bar: 50 μm. **d** Quantification of the percentage of myotubes containing ≥ 10 nuclei under each condition. **e** Quantitative analysis of average myotube diameter under indicated condition. For **d** and **e**, $N = 5$ in each group. Error bars represent s.d. *$p < 0.05$ from WT cultures transfected with vector alone by unpaired $t$-test. #$p < 0.05$ from MyD88 cultures transfected with vector alone by unpaired $t$-test. **f** Representative immunoblots presented here confirm the overexpression of p52, RelB, and IKKα proteins in transfected cultures

We found that overexpression of MyD88 significantly increased myoblast fusion in WT cultures. Furthermore, forced expression of MyD88 rescued fusion defects in MyD88-KO cultures (Fig. 3a, b).

Myoblast fusion requires several cell adhesion and transmembrane proteins, which accumulate at contact sites between two myogenic cells either in an asymmetrical or symmetrical fashion[3], [42]. Moreover, a few cytokines, such as IL-4 and IL-6, have been shown to promote myoblast fusion[30, 43]. To understand the role of MyD88 in gene expression of profusion molecules, WT and MyD88-KO myoblasts were incubated in DM for 48 h and

relative mRNA levels of β1-integrin, β1D-integrin, Adam12, myomaker, M-cadherin, N-cadherin, myoferlin, nephronectin, MOR23, caveolin-3, IL-4, and IL-6 were measured by performing quantitative real-time PCR (QRT–PCR). Interestingly, mRNA levels of β1-integrin, β1D-integrin, Adam12, myomaker, caveolin-3, and IL-6 were significantly reduced in MyD88-KO cultures compared with WT cultures (Fig. 3c). We also investigated whether the overexpression of MyD88 can induce the mRNA levels of some of these fusion-related molecules in WT myoblasts. Overexpression of MyD88 significantly increased the mRNA levels of β1D-integrin, Adam12, myomaker, myoferlin, and

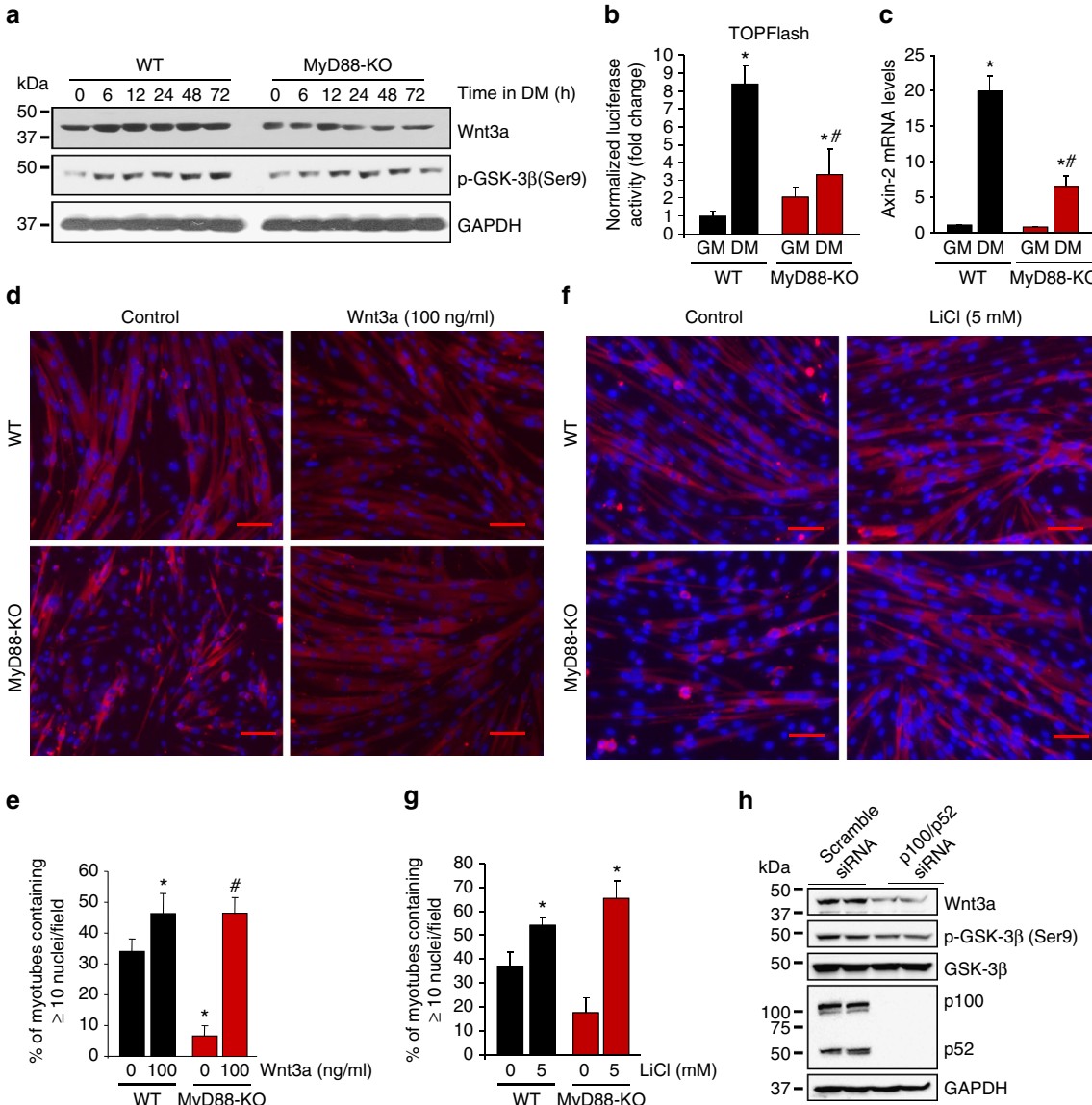

**Fig. 5** MyD88 mediates myoblast fusion through canonical Wnt pathway. **a** Primary myoblasts of WT and MyD88-KO mice were plated at equal densities, incubated in DM, and samples were collected at indicated time points. Immunoblots demonstrate the levels of Wnt3a, p-GSK-3β, and unrelated protein GAPDH. **b** WT and MyD88-KO myoblasts were co-transfected with Super (8X) TOPFlash reporter plasmid and Renilla luciferase plasmid (1:10 ratio) and were incubated in growth medium (GM) or DM for 48 h. The cells were then collected and luciferase activity was measured. **c** WT and MyD88-KO myoblasts were incubated in GM or DM for 48 h and mRNA levels of Axin-2 were measured by performing QRT–PCR assay. For **b** and **c**, $N = 4$ in each group. *$p < 0.05$ from WT cultures incubated in GM. #$p < 0.05$ from WT cultures incubated in DM. **d** WT and MyD88-KO primary myoblasts were plated at equal densities and incubated with DM with vehicle alone or with 100 ng/ml Wnt3a protein. The cultures were fixed and stained with anti-MyHC and DAPI. Representative photomicrographs are presented here. Scale bar: 50 μm. **e** Quantification of the percentage of myotubes containing ≥ 10 nuclei in control and Wnt3a-treated WT and MyD88-KO cultures. $N = 4$ in each group. *$p < 0.05$ from WT cultures without Wnt3a by unpaired $t$-test. #$p < 0.05$ from MyD88-KO cultures without Wnt3a protein by unpaired $t$-test. **f** WT and MyD88-KO primary myoblasts were plated at equal densities and incubated with DM with vehicle alone or 5 mM LiCl for 48 h. The cultures were fixed and stained with anti-MyHC and DAPI. Representative photomicrographs are presented here. Scale bar: 50 μm. **g** Quantification of the percentage of myotubes containing ≥ 10 nuclei in vehicle alone and LiCl-treated WT and MyD88-KO cultures. $N = 4$ in each group. Error bars represent s.d. *$p < 0.05$ from WT cultures without LiCl by unpaired $t$-test. #$p < 0.05$ from MyD88-KO cultures without LiCl by unpaired $t$-test. **h** Primary WT myoblasts were transfected with scrambled siRNA or p100/p52 siRNA. After 36 h, the cells were incubated in DM for 48 h followed by analysis by Western blot. Immunoblots presented here demonstrate the levels of Wnt3a, p-GSK-3β, GSK-3β, p100/p52, and unrelated protein GAPDH

caveolin-3 in myoblasts incubated in DM (Fig. 3d). Over-expression of MyD88 had no effect on MyoD levels, however, a modest increase in the levels of myogenin and MyHC was noticeable which could be an indirect effect of the activation of other MyD88-mediated signaling pathways under supra physiological conditions (Fig. 3e).

We also investigated whether overexpression of MyD88 can promote myoblast fusion in growth conditions or in non-myogenic cells. Primary myoblasts were transfected with empty vector or MyD88 cDNA and the cells were incubated in growth medium (GM) for 96 h followed by staining with May-Grünwald solution. We found that overexpression of MyD88 did not induce

myoblast fusion in GM (Supplementary Fig. 8A). In another experiment, primary WT fibroblasts were transfected with vector alone or MyD88 cDNA. The cells were then incubated in DM for 72 h and analyzed by staining with May-Grünwald solution. There was no multinucleated fibroblast in control or MyD88-overexpressing cultures (Supplementary Fig. 8B). These results suggest that MyD88 promotes fusion only in the presence of the sequentially expressed conventional MRFs during myogenic differentiation.

Since MyD88 is an important component of IL-1 receptor and TLR-mediated signaling, we next sought to investigate whether IL-1β or various TLR agonists can augment fusion of cultured myoblasts. Treatment of WT myoblasts with the IL-1β, TLR1/2 agonist Pam3CSK4, TLR2 agonist HKLM, TLR3 agonists Poly(I:C) (high molecular weight) or Poly(I:C) (low molecular weight), TLR4 agonist lipopolysaccharide (LPS), TLR5 agonist FLA-ST (Flagellin from *S. typhimurium*), TLR6/2 agonist FSL1 (Pam2CGDPKHPKSF), TLR7/8 agonist ssRNA40, or

TLR9 agonist ODN1826 had no positive effect on myoblast fusion. Indeed, a few TLR agonists such as HKLM, FLA-ST, FSL1, ssRNA40, and ODN significantly reduced fusion of cultured myoblasts (Supplementary Fig. 9A).

Endogenous MyD88 protein is present as small speckle-like condensed forms within cytoplasm portion of cells. In response to TLR stimulation, aggregation of MyD88 is increased, which may be required for the activation of downstream pathways[37, 44]. We found that MyD88 aggregates were scantly scattered in the cytoplasm of undifferentiated myoblasts. Treatment of myoblasts with LPS led to the formation of more MyD88 aggregates. While the amount of MyD88 was increased, there was no evidence of MyD88 aggregate formation upon incubation of myoblasts in DM (Supplementary Fig. 9B) further suggesting that MyD88 promotes myoblast fusion in a cell autonomous manner and independent of signaling from IL-1 receptor or TLRs.

**MyD88 regulates non-canonical NF-κB during myogenesis**. We next sought to determine whether MyD88 regulates profusion signaling pathways during myogenesis. There was no significant difference in the levels of phospho-ERK1/2, phospho-FAK, and phospho-ERK5 between WT and MyD88-KO cultures. Furthermore, there was no difference in the levels of NFATc2 between WT and MyD88-KO cultures at different time points after addition of DM (Supplementary Fig. 10), suggesting that MyD88 does not affect the activation of ERK1/2, FAK, ERK5, and calcineurin/NFATc2 during myogenic differentiation. Interestingly, we found that the proteolytic processing of the p100 into p52, a marker for the activation of non-canonical NF-κB pathway[22, 23], was considerably inhibited in MyD88-KO cultures compared to WT cultures after addition of DM (Fig. 4a). There was no difference in the levels of IκBα, a marker for the activation of canonical NF-κB pathway[22, 23], between WT and MyD88-KO cultures suggesting that MyD88 specifically mediates the activation of the non-canonical NF-κB pathway during myogenesis (Fig. 4a). To further understand the role of MyD88 in activation of non-canonical NF-κB signaling during myogenesis, we transfected primary myoblasts with empty vector or MyD88 cDNA and incubated them in DM for 48 h followed by measuring the levels of p100 and p52 proteins. Results showed that overexpression of MyD88 increased the levels of p52 in myogenic cultures (Fig. 4b).

We next sought to determine whether the inhibition of non-canonical NF-κB signaling is a mechanism for myoblast fusion deficiency in MyD88-KO cultures. Consistent with a published report[26], we found that overexpression of components of non-canonical NF-κB pathway: p52, RelB, or IKKα signcantly

increased myoblast fusion in WT cultures. Furthermore, over-expression of p52, RelB, or IKKα also improved myoblast fusion in MyD88-KO cultures (Fig. 4c–f). These results suggest that MyD88 mediates myoblast fusion through the activation of the non-canonical NF-κB signaling.

**MyD88 activates canonical Wnt signaling during myogenesis**. The canonical Wnt pathway has been shown to induce myoblast fusion both in vitro and in vivo[14, 21]. By performing Western blotting, we first compared the levels of Wnt3a in WT and MyD88-KO myoblast cultures at different time points after addition of DM. The levels of Wnt3a were considerably increased in WT cultures after addition of DM. In contrast, there was no such increase in the levels of Wnt3a in MyD88-KO cultures (Fig. 5a). Phosphorylation of GSK-3β at Ser9 leads to the inactivation of its enzymatic activity, a step which is important for the activation of canonical Wnt pathway[19]. Our results showed that the levels of phosphorylated GSK-3β (Ser9) were reduced in MyD88-KO cultures compared to WT cultures during myogenic differentiation (Fig. 5a). We also studied the activation of canonical Wnt signaling using the TOPFlash reporter system, which measures TCF/LEF-dependent transcriptional activity[21]. As shown in Fig. 5b, luciferase activity was significantly reduced in MyD88-KO cultures compared with WT cultures upon incubation in DM. Moreover, mRNA levels of Axin-2, a target gene of canonical Wnt signaling, were significantly reduced in MyD88-KO cultures compared with WT cultures (Fig. 5c) confirming inhibition of canonical Wnt signaling in MyD88-KO cultures upon induction of the differentiation program.

We next investigated whether forced activation of canonical Wnt signaling can rescue fusion defects in MyD88-KO cultures. Interestingly, addition of recombinant Wnt3a protein dramatically improved myotube formation in MyD88-KO cultures (Fig. 5d, e). LiCl activates canonical Wnt signaling through inhibiting GSK-3β[45]. Similar to Wnt3a, we found that LiCl also significantly improved myotube formation in MyD88-KO cultures (Fig. 5f, g).

Since both non-canonical NF-κB and canonical Wnt signaling mediate myoblast fusion, we also investigated whether MyD88-mediated activation of non-canonical NF-κB pathway contributes to the activation of canonical Wnt signaling during myogenesis. Primary WT myoblasts were transfected with scrambled siRNA or p100/p52 siRNA followed by incubation in DM for 48 h. Interestingly, knockdown of p100/p52 considerably reduced the levels of Wnt3a and phosphorylation of GSK-3β (at Ser9) in myogenic cultures (Fig. 5h). These results suggest that downstream of MyD88, non-canonical NF-κB signaling contributes to the activation of canonical Wnt signaling during myogenesis.

**MyD88 mediates myoblast fusion during muscle regeneration**. Skeletal muscle regeneration involves the activation and proliferation of satellite cells, which eventually differentiate into myoblasts and fuse with injured myofibers to accomplish regeneration[46–48]. We first studied how the levels of MyD88 are regulated in response to skeletal muscle injury. TA muscle of 12-week-old WT mice was given an intramuscular injection of 1.2% BaCl₂ to induce muscle injury while the contralateral muscle was injected with saline and served as a control. After 3 days, the muscles were collected and analyzed by QRT–PCR and western blotting. Results showed that both mRNA and proteins levels of MyD88 were dramatically increased in injured skeletal muscle compared with control TA muscle (Supplementary Fig. 11A, 11B). We next sought to compare skeletal muscle regeneration between 12-week-old WT and MyD88-KO mice. There was no overt difference in TA muscle regeneration between WT and

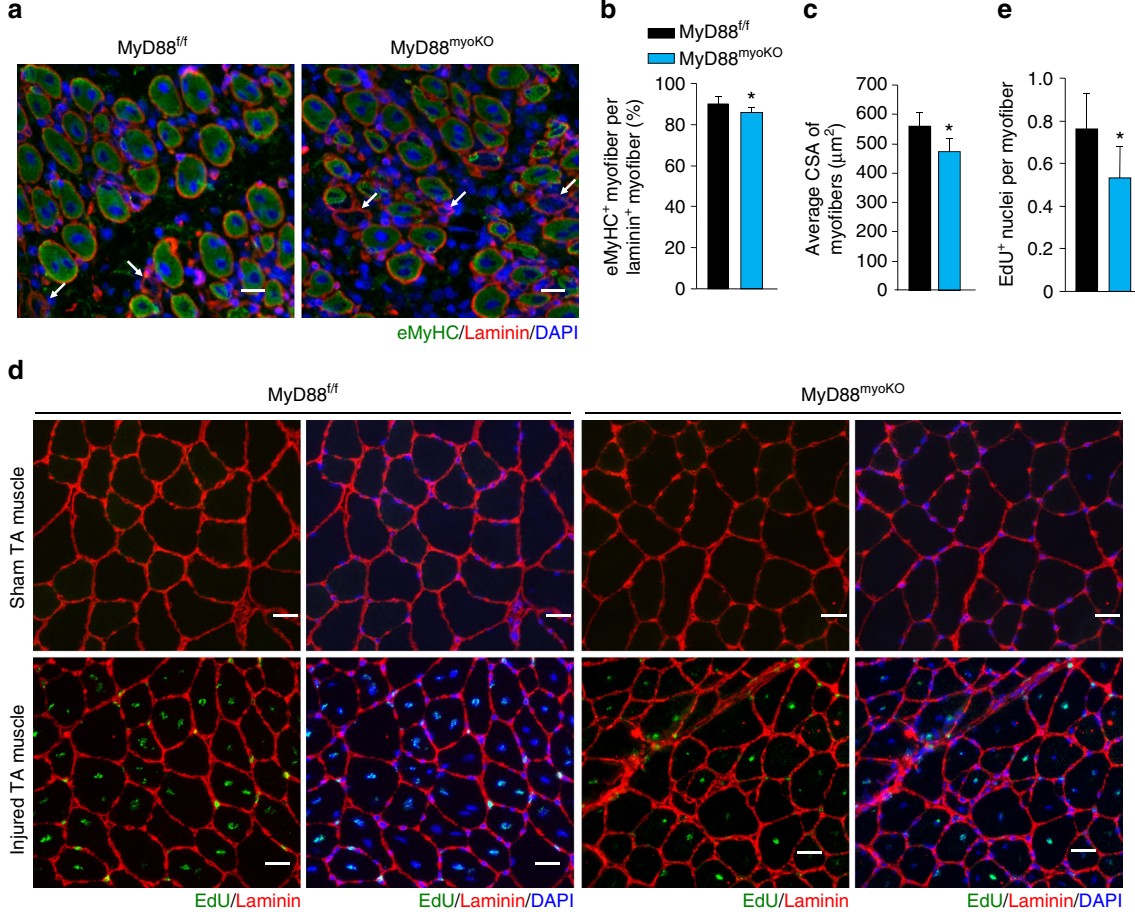

**Fig. 6** MyD88 mediates myoblast fusion during skeletal muscle regeneration. **a** TA muscle of 12-week-old MyD88[f/f] and MyD88[myoKO] mice was injured by intramuscular injection of 100 μl of 1.2% BaCl₂ solution. After 5 days, the TA muscle was isolated and immunostained for eMyHC (*green*) and laminin (*red*) proteins. Nuclei were counterstained using DAPI (*blue*). Representative images are presented here. Arrows point to eMyHC⁻/laminin⁺ fibers. Scale bar: 50 μm. **b** Quantification of percentage of eMyHC⁺ myofibers within laminin staining, and **c** average CSA of eMyHC⁺ myofiber in TA muscle of MyD88[f/f] and MyD88[myoKO] mice. $N = 3$ in each group. *$p < 0.05$ from MyD88[f/f] mice by unpaired $t$-test. In another experiment, TA muscle of MyD88[f/f] and MyD88[myoKO] mice was injured by intramuscular injection of 100 μl of 1.2% BaCl₂ solution. After 2 days, the mice were given an intraperitoneal injection of EdU and 11 days later TA muscles were collected and muscle sections prepared were stained to detect EdU, laminin, and nuclei. **d** Representative photomicrographs after EdU, laminin, and DAPI staining are presented here. Scale bar: 50 μm. **e** Quantification of the percentage of EdU⁺ nuclei per muscle fiber. $N = 5$ in each group. Error bars represent s.d. *$p < 0.01$ from MyD88[f/f] mice by unpaired $t$-test

MyD88-KO mice at day 5 post-injury (Supplementary Fig. 11C). However, average CSA of centronucleated myofibers was significantly reduced in TA muscle of MyD88-KO mice compared with WT mice (Supplementary Fig. 11D, 11E).

To investigate the cell-autonomous role of MyD88 in regenerative myogenesis, TA muscle of 12-week-old MyD88[f/f] and MyD88[myoKO] mice was injured by an intramuscular injection of 1.2% BaCl₂ solution. After 5 days, the TA muscle was isolated and analyzed by performing immunostaining for embryonic myosin heavy chain (eMyHC) and laminin (Fig. 6a). Results showed that the number of eMyHC⁺ myofibers within laminin staining and average CSA of eMyHC⁺ myofibers were significantly reduced in TA muscle of MyD88[myoKO] mice compared to MyD88[f/f] mice (Fig. 6b, c). However, there was no significant difference in the percentage of satellite cells or inflammatory immune cells in 5d-injured TA muscle of MyD88[f/f] and MyD88[myoKO] mice (Supplementary Fig. 12).

To evaluate whether the deficit in muscle regeneration in MyD88[myoKO] mice is attributed to the reduced fusion capacity of myoblasts, TA muscle of 12-week-old MyD88[f/f] and MyD88[myoKO] mice was injured by an intramuscular injection of 1.2% BaCl₂ solution. After 48 h, the mice were given intraperitoneal

injection of EdU and the TA muscle was isolated 11 days later, and stained for the detection of EdU⁺ myonuclei (Fig. 6d). This analysis showed that the number of EdU⁺ nuclei per myofiber was significantly reduced in MyD88[myoKO] mice compared with corresponding MyD88[f/f] mice confirming that ablation of MyD88 diminishes fusion of myoblasts to injured myofibers during regenerative myogenesis (Fig. 6e).

**MyD88 promotes fusion of transplanted myoblasts to myofibers.** We also investigated the effect of overexpression of MyD88 on myoblast fusion in vivo. Primary myoblasts prepared from the hind limb muscle of mTmG reporter mice were stably transfected with empty vector or MyD88 cDNA. TA muscle of WT mice was injured by intramuscular injection of 1.2% BaCl₂ solution. After 24 h, empty vector- or MyD88-overexpressing mT myoblasts were injected into the injured TA muscle. The TA muscle was isolated at 28d post-myoblast injection and muscle sections generated were analyzed using a fluorescence microscope. Transplanted mT⁺ myoblasts fused with injured myofibers, as evidenced by red fluorescence (Fig. 7a). Although not statistically significant, there was a trend toward reduced number of mT⁺ myofibers in TA muscle transplanted with MyD88-

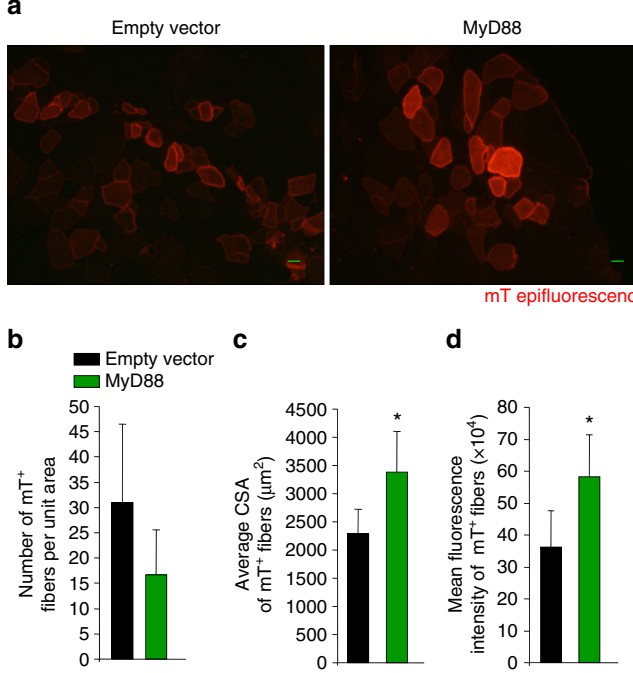

**Fig. 7** Overexpression of MyD88 promotes fusion of transplanted myoblasts to injured myofibers in vivo. **a** Primary myoblasts were prepared from hind limb muscle of mTmG mice and transfected with empty vector or vector containing MyD88 cDNA. Cells were then transplanted in TA muscle of WT mice that have been subjected to BaCl₂-mediated injury for 24 h. After 28 days, the TA muscles were isolated and processed for histological analysis. Representative photomicrographs depicting mT⁺ myofibers originating from empty vector or MyD88-transfected transplanted myoblasts. Scale bar: 20 μm. **b** Quantification of number of mT⁺ engrafted myofibers per unit area. **c** Quantification of average CSA of mT⁺ engrafted myofibers in TA muscle sections. **d** Quantification of mean fluorescence intensity of mT⁺ myofibers in both groups. $N = 4$ in each group. Error bars represent s.d. *$p < 0.05$ from empty vector by unpaired $t$-test

overexpressing myoblasts compared to those transplanted with control myoblasts (Fig. 7b). However, the average CSA of mT⁺ myofibers and mean florescence intensity were significantly increased in the TA muscle transplanted with MyD88-overexpressing mT⁺ myoblasts compared to corresponding TA muscle transplanted with control mT⁺ myoblasts (Fig. 7c, d). Altogether, these results suggest that overexpression of MyD88 promotes fusion of myoblasts with injured myofibers in vivo.

**MyD88 mediates overload-induced muscle hypertrophy**. Myoblast fusion is essential for skeletal muscle hypertrophy and hyperplasia in response to functional overload[5, 43, 49]. To further understand the physiological significance of MyD88 in myoblast fusion, we studied the effect of myoblast-specific deletion of MyD88 in overload-induced muscle hypertrophy. For this experiment, we used a bilateral synergistic ablation model to induce hypertrophy of the plantaris muscle of adult mice[43, 50, 51]. We found that the levels of MyD88 were considerably increased in plantaris muscle of adult WT mice after 14 days of functional overload (Fig. 8a). We next investigated the effect of myoblast-specific deletion of MyD88 on overload-induced skeletal muscle hypertrophy. Interestingly, overload-induced myofiber hypertrophy in plantaris muscle was significantly reduced in MyD88^myoKO mice compared with MyD88^f/f mice (Fig. 8b, c). Moreover, overload-induced increase in number of myofibers (hyperplasia)

was also significantly reduced in MyD88^myoKO mice (Fig. 8d). To specifically investigate whether reduced myofiber hypertrophy in MyD88^myoKO mice in response to functional overload is attributed to reduced myoblast fusion, we performed an EdU incorporation assay. Twelve-week-old littermate MyD88^f/f and MyD88^myoKO mice were subjected to sham or bilateral synergistic ablation surgery. After 48 h, the mice were given an intraperitoneal injection of EdU and 12 days later, the plantaris muscle was isolated and processed for the detection of EdU (Fig. 8e). We found that ~25% of myofibers in MyD88^f/f mice contained EdU⁺ nuclei after 14 days of synergistic ablation surgery. Importantly, the percentage of myofibers containing EdU⁺ nuclei was significantly reduced in MyD88^myoKO mice compared to corresponding MyD88^f/f mice (Fig. 8f). These results further suggest that MyD88 promotes skeletal muscle growth in adult mice through augmenting myoblast fusion.

## Discussion

MyD88 is a ubiquitously expressed adaptor protein which regulates the proliferation and differentiation of the cells involved in the innate immune response[36]. Although a few studies have previously investigated the role of MyD88 in skeletal muscle, they were focused toward understanding the role of TLRs and associated inflammatory response in models of chronic muscle injuries such as muscular dystrophy[52–54] or ischemic injury[53, 55], which involves divergent inflammatory response and myopathy[56].

In this study, we have discovered that MyD88 levels are upregulated during myogenesis and that it promotes skeletal muscle growth in multiple conditions including postnatal development through mediating myoblast fusion. It is well-known that MyD88 is an important mediator for the activation of downstream signaling pathways in response to TLRs and IL-1 receptor[37, 39, 57]. Intriguingly, we found that agonists of TLRs or IL-1β do not augment myoblast fusion, implying that MyD88 promotes myoblast fusion independent of receptor-mediated signaling (Supplementary Fig. 9). The cell autonomous role of MyD88 in myoblast fusion is also evidenced by the findings that even though the cytoplasmic levels of MyD88 were increased after induction of myogenesis, there was no evidence of MyD88 aggregates which was noticeable after treatment of cells with TLR4 agonist, LPS (Supplementary Fig. 9). Our findings are consistent with recent studies that suggest that MyD88 can regulate tissue homeostasis in a cell autonomous manner[39]. For example, MyD88 is required in a cell autonomous fashion for RAS-mediated tumorigenesis[58]. MYD88 mutations have been implicated in a number of human malignancies, including Waldenström's macroglobulinemia[39]. A recent study investigated the effects of single MYD88 mutation in otherwise normal mature B cells and found that MYD88 L265P mutation alone is sufficient to cause mitogen-independent B cell proliferation[59].

Myoblast fusion is a highly coordinated process which involves the activation of several signaling pathways and transcription factors that augment levels of specific molecules which accumulate at contact sites between two myogenic cells to facilitate fusion[3, 9, 14]. Our results show that one of the important mechanisms by which MyD88 promotes myoblast fusion is through augmenting the expression of genes whose products are involved in fusion. In fact, our loss-of-function and gain-of-function experiments revealed that β-integrin, β1D-intergrin, Adam12, myomaker, myoferlin, caveolin-3, and IL-6 are some of the important profusion molecules that are regulated through MyD88-dependent mechanisms during myogenesis (Fig. 3).

The NF-κB transcription factor can be activated through canonical or non-canonical pathways[23]. During myogenesis, the

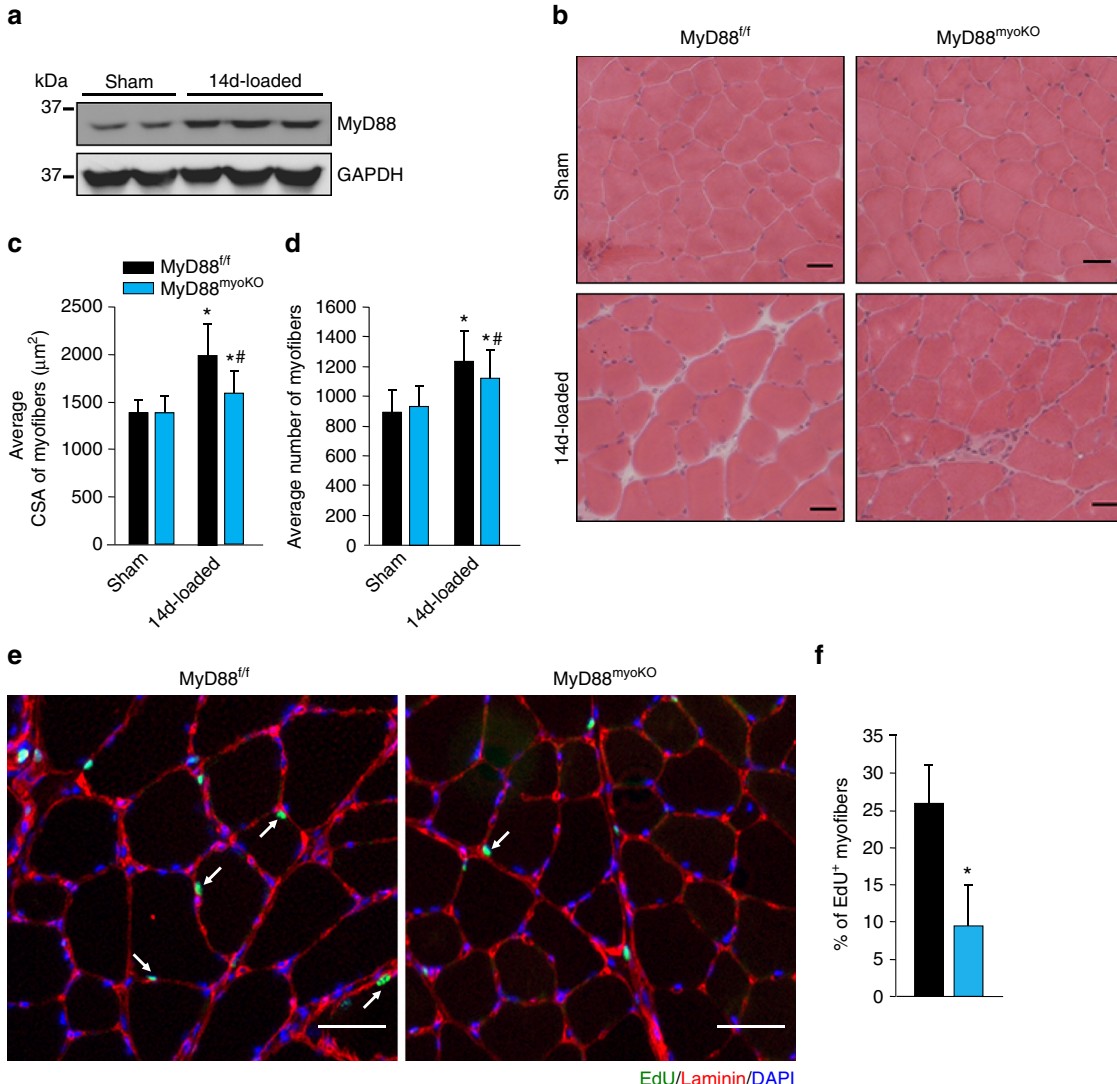

**Fig. 8** MyD88 promotes over load-induced myofiber hypertrophy in adult mice. **a** Twelve-week-old WT mice were subjected to sham or synergistic ablation surgery for 14d and the plantaris muscle was analyzed for the protein levels of MyD88 and unrelated protein GAPDH. **b** Twelve-week-old MyD88$^{f/f}$ and MyD88$^{myoKO}$ mice were subjected to sham or synergistic ablation surgery for 14d and the plantaris muscle was isolated and stained with H&E. Representative photomicrographs of H&E-stained plantaris muscle sections are presented here. Scale bar: 50 μm. **c** Quantification of average myofiber CSA in the H&E-stained sections of plantaris muscle. **d** Average number of myofibers in plantaris muscle of MyD88$^{f/f}$ and MyD88$^{myoKO}$ mice at 14d after synergistic ablation surgery. For **c** and **d**, $N = 7$ mice in each group. In a separate experiment, 12-week-old MyD88$^{f/f}$ and MyD88$^{myoKO}$ mice were subjected to sham or synergistic ablation surgery. After 2 days, the mice were given an intraperitoneal injection of EdU, and 12 days later plantaris muscles were collected, and muscle sections prepared were stained to detect EdU, laminin, and nuclei. **e** Representative photomicrographs after EdU, laminin, and DAPI staining are presented here. Arrows point to intrafiber EdU$^+$ nuclei. Scale bar: 100 μm. **f** Quantification of percentage myofibers containing EdU$^+$ nuclei in plantaris muscle of MyD88$^{f/f}$ and MyD88$^{myoKO}$ mice. $N = 4$ in each group. Error bars represent s.d. *$p < 0.05$ from corresponding sham-operated MyD88$^{f/f}$ or MyD88$^{myoKO}$ mice by unpaired $t$-test. #$p < 0.05$ from 14d-loaded muscle of MyD88$^{f/f}$ mice by unpaired $t$-test

canonical NF-κB pathway is repressed, whereas the non-canonical NF-κB pathway gets upregulated[24, 25]. Recently, it was reported that the non-canonical NF-κB pathway also mediates myoblast fusion during myogenesis[26]. Our results suggest that MyD88 promotes myoblast fusion, at least in part, through the activation of non-canonical NF-κB signaling (Fig. 4). While there is a plethora of literature demonstrating that MyD88 mediates the activation of the canonical arm of NF-κB in response to IL-1β and agonists of TLRs[37, 60–62], there is limited evidence about the role of MyD88 in the activation of non-canonical NF-κB pathway. It has been reported that LPS stimulates the phosphorylation of NF-κB-inducing kinase (NIK), which phosphorylates IKKα, the major regulator of non-canonical NF-κB pathway[63]. However, it remains unknown whether NIK is

recruited to TLR4 or to the MyD88 signaling complex. Our results suggest that MyD88 promotes myoblast fusion during myogenesis independent of TLR-mediated signaling (Supplementary Fig. 9A). It has been consistently observed that TRAF6 is an important component of TLRs-MyD88 signaling in many cell types[36]. While TRAF6 is required for myogenic differentiation, it does not specifically mediate myoblast fusion[47] further suggesting that MyD88 regulates myoblast fusion in a cell autonomous manner and independent of receptor-mediated signaling.

Wnt signaling also plays an important role in myogenesis including myoblast fusion[19]. Upon binding to their receptors, Wnt proteins induce a cascade of intracellular signaling events, which culminate in the stabilization of β-catenin. β-catenin translocates to the nucleus, where it binds to T-cell factor/

lymphoid enhancer-binding factor (TCF/LEF) to induce the transcription of Wnt target genes[64]. GSK-3β is a critical component of the classical Wnt signaling pathway because its inactivation leads to the stabilization and nuclear translocation of β-catenin[19]. Inhibition of GSK-3β using LiCl increases nuclear accretion and diameter of cultured myotubes[20]. Similarly, Wnt3a protein augments fusion in cultured myoblasts[65]. Moreover, an intramuscular injection of Wnt3a protein resulted in an increased size of regenerating myofibers without having any effect on myofiber number[21]. Our results show that MyD88 is involved in the activation of canonical Wnt signaling during myogenesis (Fig. 5a–c) and forced activation of canonical Wnt signaling rescues fusion defects in MyD88-KO cultures (Fig. 5d–g).

While our results suggest that MyD88 regulates the non-canonical NF-κB and the canonical Wnt pathways during myogenesis, the molecular interactions through which these two signaling cascades are regulated through MyD88 remain enigmatic. It is possible that MyD88 interacts with specific components of these two signaling pathways leading to their activation. Indeed, it has been reported that in some conditions, MyD88 interacts with TRAF3, an adaptor protein which also regulates the activation of non-canonical NF-κB signaling pathway[66]. Although more investigations are needed to determine whether MyD88 activates canonical Wnt and non-canonical NF-κB pathways independently or through signaling cross-talk, our results demonstrate that siRNA-mediated knockdown of p100/p52 subunit of NF-κB significantly reduced the levels of Wnt3a protein and phosphorylation of GSK-3β (Ser9) during myogenesis (Fig. 5h). Therefore, it is likely that non-canonical NF-κB signaling pathway induces the expression of certain Wnt ligands, such as Wnt3a, which stimulate canonical Wnt signaling and vice versa.

In addition to postnatal growth and in vitro myogenesis, we found that MyD88 also mediates myoblast fusion in skeletal muscle of adult mice. Although ablation of MyD88 reduces myoblast fusion, it does not completely blunt this process in vivo. The milder phenotype in vivo could be attributed to the fact that reduced level of Wnt ligands such as Wnt3a observed in MyD88-KO cultures is compensated for through their production by other cell types present in the muscle microenvironment during postnatal muscle growth, regeneration, and overload-induced myofiber hypertrophy. Similarly, it is possible that the activation of non-canonical NF-κB signaling pathway is mediated by other cytokines and growth factors that may be present in skeletal muscle in vivo. There is also a possibility that the activation of other signaling pathways compensate for the inhibition of canonical Wnt and non-canonical NF-κB signaling in skeletal muscle of MyD88myoKO mice. This possibility is also supported by published reports demonstrating that the inhibition of some other profusion pathways reduces the size of regenerating myofibers but does not completely blunt skeletal muscle regeneration[14, 19, 29, 30].

In summary, the present study provides initial evidence that MyD88 mediates myoblast fusion during myogenesis in a cell autonomous manner. Although more investigations are needed, our findings suggest that enhancing the levels of MyD88 in muscle progenitor cells can be a therapeutic approach for various muscle degenerative disorders and a means by which engraftment of exogenous myoblasts in cellular therapies can be enhanced.

## Methods

**Animals**. MyD88-KO (Jax strain: B6.129P2 (SJL)-MyD88tm1.1Defr/J), floxed MyD88 (MyD88f/f; Jax strain: B6.129P2 (SJL)-MyD88tm1Defr/J), and mTmG (Jax strain: Gt(ROSA)26Sortm4(ACTB–tdTomato,–EGFP)Luo/J) mice were purchased from Jackson Laboratory. Myoblast-specific MyD88-knockout (MyD88myoKO) mice were generated by crossing MyD88f/f mice with Myod1-Cre (Jax strain: FVB.Cg-Myod1tm2.1(icre)Glh/J) mice. All mice were in the C57BL6 background and their genotype was determined by PCR from tail DNA. All experimental protocols with

mice were approved in advance by the Institutional Animal Care and Use Committee (IACUC) and Institutional Biosafety Committee at the University of Louisville (IACUC numbers: 13097 and 16663).

**Skeletal muscle injury and in vivo fusion assay**. At 12 weeks of age, 100 µl of 1.2% BaCl$_2$ (Sigma Chemical Co.) dissolved in saline was injected into the TA muscle of male mice to induce necrotic injury as described[38, 41, 47]. At various time points, TA muscle was collected from killed mice for biochemical and histological studies. In one experiment, $0.5 \times 10^6$ mT myoblasts transfected with empty vector or vector containing MyD88 cDNA were injected into the TA muscle of mice 24 h after 1.2% BaCl$_2$-mediated injury. To study myoblast fusion in vivo, male MyD88f/f and MyD88myoKO mice were given an intraperitoneal injection of EdU (4 µg per gram body weight) at day 2 after intramuscular injection of 1.2% BaCl$_2$ into the TA muscle. After 11 days of EdU injection, the TA muscle was isolated and sectioned in a microtome cryostat. For studying myoblast fusion during postnatal growth period, mice were injected with EdU at P5 and P8 and killed at the age of 2 weeks. Transverse muscle sections made were immunostained with anti-laminin and for detection of EdU and nuclei. The number of intrafiber EdU$^+$ myonuclei/myofiber was quantified using NIH ImageJ software.

**Synergistic ablation surgery**. The mice were subjected to bilateral synergistic ablation surgery to induce myofiber hypertrophy of the plantaris muscle as described[67]. In brief, male mice were anesthetized using tribromoethanol and the soleus and ~60% of the gastrocnemius muscles were surgically excised while ensuring that the neural and vascular supply remained intact and undamaged for the remaining plantaris muscle. A sham surgery was performed for controls following exactly the same procedures except that gastrocnemius and soleus muscles were not excised. After 14 days, the muscle was isolated and processed for morphometric and biochemical analysis. To study the contribution of myoblast fusion in overload-induced hypertrophy, in a separate experiment, 2 days after synergistic ablation surgery, MyD88f/f and MyD88myoKO mice were given an intraperitoneal injection of EdU (4 µg/g body weight). After 12 days of EdU injection, the plantaris muscle was isolated and transverse muscle sections made were immunostained with anti-laminin and for detection of EdU and nuclei. The number of intrafiber EdU$^+$ myonuclei/myofiber was quantified using NIH ImageJ software. Samples were blinded for analysis.

**Immunohistochemistry**. For immunohistochemistry studies, frozen TA or plantaris muscle sections were fixed in acetone or 4% paraformaldehyde (PFA) in PBS, blocked in 2% bovine serum albumin in PBS for 1 h and incubated with rabbit anti-dystrophin, mouse anti-Pax7, mouse anti-eMyHC, rabbit anti-laminin, in blocking solution at 4 °C overnight under humidified conditions. The sections were washed briefly with PBS before incubation with goat anti-rabbit Alexa Fluor 468, goat anti-mouse Alexa Fluor 594, goat anti-rabbit Alexa Fluor 488 or goat anti-mouse Alexa Fluor 488 secondary antibody for 1 h at room temperature and then washed three times for 15 min with PBS. Nuclei were counterstained with DAPI. The slides were mounted using fluorescence medium (Vector Laboratories) and visualized at room temperature on Nikon Eclipse TE 2000-U microscope (Nikon), a digital camera (Nikon Digital Sight DS-Fi1), and NIS Elements BR 3.00 software (Nikon). Image levels were equally adjusted using Photoshop CS6 software (Adobe).

**Histology and morphometric analysis**. TA or plantaris muscle of mice was removed, frozen in isopentane cooled in liquid nitrogen, and sectioned in a microtome cryostat. For the assessment of tissue morphology, 10 µm-thick transverse sections of muscles were prepared and stained with Hematoxylin and Eosin (H&E). The sections were examined under an inverted microscope (Nikon Eclipse TE 2000-U) using a Plan 10x, NA 0.25 PH1 DL or Plan-Fluor ELWD 20x, NA 0.45 Ph1 DM objective lens, a digital camera (Digital Sight DS-Fi1) and NIS Elements BR 3.00 software (Nikon). Images of H&E-stained or dystrophin-stained TA or plantaris muscle sections were analyzed by measuring myofiber cross-sectional area (CSA). A minimum of 90% centrally nucleated myofibers taken from the center of the injured muscle region was used in the analysis. The distribution of myofiber CSA was calculated by analyzing 200–250 myofibers per field using the NIS Elements BR 3.00 software (Nikon) as described[47, 48].

**Primary myoblast and fibroblast cultures**. Primary myoblasts and fibroblasts were isolated from the hind limbs of 8-week-old male or female mice[47]. Briefly, WT and/or MyD88-KO mice were killed and the hind limb muscles were isolated and excess connective tissues and fat were cleaned in sterile PBS. Muscle tissues were then minced into a coarse slurry and enzymatically digested at 37 °C for 1 h by adding 400 IU/ml collagenase II (Worthington). The digested slurry was spun, pelleted, and triturated multiple times, and then sequentially passed through a 70-µm and then 30-µm cell strainer (BD Falcon). The filtrate was spun at $1000 \times g$ and suspended in myoblast growth medium (Ham's F-10 medium with 20% FBS supplemented with 10 ng/ml of basic fibroblast growth factor). Cells were first re-fed after 3 days of initial plating. Cells were pre-plated for 15–30 min for the first few passages to select for a pure myoblast population (cells in suspension) or pure fibroblast population (adherent cells). Upon selection of each cell type, the cells were cultured in their corresponding culturing medium till reaching 80%

confluence. To induce differentiation, the cells were incubated in differentiation medium (DM; 2% horse serum in DMEM).

**Isolation of single myofibers.** Single myofiber cultures were established following a protocol as described[47, 68]. Extensor digitorum longus muscle was isolated tendon to tendon from mice. The muscle was then enzymatically digested in DMEM medium supplemented with collagenase (400 IU/ml, Worthington Biochemical Corporation, Lakewood, NJ) at 37 °C for 45 min. Single myofibers were then released by trituration in myofiber culture medium (DMEM with 10% fetal bovine serum (FBS)). Immediately after isolation, myofibers were fixed with 4% PFA in PBS followed by staining with DAPI to detect myonuclei. The number of DAPI-stained nuclei per unit myofiber length was determined by manual counting.

**Lactate dehydrogenase assay.** The amount of lactate dehydrogenase (LDH) in culture supernatants was measured using a LDH Cytotoxicity Assay kit and following the protocol suggested by the manufacturer (Thermo Scientific).

**Plasmids and gene transfer by electroporation.** MyD88 flag plasmid was a gift from Ruslan Medzhitov (Addgene plasmid # 13093). RelB cFlag pcDNA3 (Addgene plasmid # 20017) and p52 cFlag pcDNA3 (Addgene plasmid # 20019) were gifts from Stephen Smale. pcDNA3-IKKα-HA WT was a gift from Warner Greene (Addgene plasmid # 23296). Plasmid DNA was electroporated into primary myoblasts (1500 V, 10 ms duration, 3 pulses) or into primary fibroblast (900 V, 40 ms duration, 1 pulse) using the Neon transfection system as previously described[41, 47]. This protocol consistently introduced plasmid DNA into ~80% cells.

**Transient transfection and reporter gene activity.** Primary myoblasts were seeded on 6-well cell culture plates coated with 10% matrigel (BD Biosciences) and transfected with scrambled siRNA, MyD88 siRNA, or p100/p52 siRNA oligonucleotides (Santa Cruz Biotechnology) using Lipofectamine RNAiMAX Transfection Reagent (Invitrogen). Super (8X) TOPFlash plasmid was a gift from Randall Moon (Addgene plasmid # 12456). For reporter gene assay, the Super (8X) TOPFlash reporter plasmid was transfected in WT and MyD88-KO myoblasts using Lipofectamine 2000 following a protocol suggested by the manufacturer (Invitrogen). Transfection efficiency was controlled by co-transfection of pRL-TK Renilla luciferase reporter plasmid. After 24 h of transfection, the cells were incubated in GM or DM for 48 h and the samples were processed for luciferase expression using the Dual-luciferase assay systems with reporter lysis buffer according to the manufacturer's instructions (Promega). Luciferase measurements were made using a luminometer (Berthold Detection Systems). Data were presented as a fold change in TOPFlash luciferase activity normalized by Renilla luciferase.

**Immunocytochemistry and fusion assays.** Myotubes were fixed with 4% PFA in PBS for 15 min at room temperature and then permeabilized with 0.1% Triton X-100 in PBS for 5–8 min. Cells were blocked with 2% BSA in PBS and incubated with mouse anti-MyHC (MF-20) overnight at 4 °C and goat anti-mouse Alexa Fluor 568 at room temperature for 1 h. Cells were counterstained with DAPI for 3 min. Stained cells were photographed and analyzed using a fluorescent inverted microscope (Nikon Eclipse TE 2000-U), a digital camera (Digital Sight DS-Fi1), and Elements BR 3.00 software (Nikon). To measure fusion efficiency, more than one hundred of MyHC positive myotubes containing 2 or more nuclei were counted. More than 5 field images were analyzed per experimental group. To measure average diameter of myotubes, 100–120 myotubes per group were included. For consistency, diameters were measured at the midpoint along with the length of the MyHC⁺ myotubes. Myotube diameter was measured using the NIS Elements software. Results presented are from 4–5 independent experiments. For detection of MyD88 aggregates, myoblast were either incubated in GM in the presence of 10 μg/ml LPS or vehicle alone or incubated in DM for 48 h. Cultures were then processed for immunocytochemical analysis as mentioned above followed by incubation with goat anti-MyD88 for 3 h at room temperature and goat anti-goat Alexa Fluor 488 for 1 h. Cells were then counterstained with DAPI for 3 min. Stained cells were photographed and analyzed using a fluorescent inverted microscope (Nikon Eclipse TE 2000-U), a digital camera (Digital Sight DS-Fi1), and Elements BR 3.00 software (Nikon). Image levels were equally adjusted using Photoshop CS6 software (Adobe).

**EdU staining and proliferation assays.** EdU (5-ethynyl-2'-deoxyuridine) is a nucleoside analog of thymidine that is incorporated into DNA during cellular proliferation, which allows the visualization of newly synthesized DNA. To determine the proliferation rate of WT and MyD88-KO myoblasts, cells were counted and seeded in a 24-well tissue culture plate coated with 10% matrigel. After 24 h, 10 μM EdU (Invitrogen) was added in culture medium. The cells were incubated for 30, 60, or 90 min and the EdU⁺ cells were visualized using Click-iT EdU Cell Proliferation Assay kit (Invitrogen). Nuclei were counterstained with Hoechst dye for 3 min. Images were visualized on Nikon Eclipse TE 2000-U microscope (Nikon), a digital camera (Nikon Digital Sight DS-Fi1), and analyzed using Nikon NIS Elements BR 3.00 software (Nikon).

**Cell mixing experiment.** Primary myoblasts were isolated from hind limb muscles of WT or MyD88-KO mice and electroporated with plasmids expressing GFP or mCherry, respectively. After 36 h, equal number of WT (i.e., GFP-expressing) or MyD88-KO (i.e., mCherry-expressing) cells were plated separately or mixed at a 1:1 ratio in matrigel coated 24-well plates. Cells were then incubated in DM for 48 h and possible formation of chimeric myotubes was determined via co-expression of GFP and mCherry in the same myotube.

**Western blot.** The relative amount of various proteins was determined by performing Western blot as previously described[41, 47]. Skeletal muscle of mice or cultured myoblasts or myotubes were washed with PBS and homogenized in lysis buffer (50 mM Tris-Cl (pH 8.0), 200 mM NaCl, 50 mM NaF, 1 mM dithiothreitol, 1 mM sodium orthovanadate, 0.3% IGEPAL, and protease inhibitors). Approximately 50–100 μg protein was resolved on each lane on 10% SDS–PAGE gel, transferred onto a nitrocellulose membrane and probed using specific primary antibody at indicated dilution (Supplementary Table 1). Bound antibodies were detected by secondary antibodies conjugated to horseradish peroxidase (Cell Signaling Technology). Signal detection was performed by an enhanced chemiluminescence detection reagent (Bio-Rad). Approximate molecular masses were determined by comparison with the migration of prestained protein standards (Bio-Rad). Uncropped gel images are presented in Supplementary Fig. 13.

**RNA isolation and QRT–PCR.** Total RNA was isolated from cultured myogenic cells or skeletal muscle tissues of mice using the RNeasy Mini Kit (Qiagen). Any contaminating DNA was removed using the DNA-free kit from Ambion (Austin, TX). The quantity of RNA was analyzed using NanoDrop instrumentation (NanoDrop Technologies, Wilmington, DE). Purified RNA (1–4 μg) was used to synthesize first strand of cDNA by reverse transcription system using Ambion's oligo-dT primer and Qiagen's Omniscript reverse transcriptase according to the manufacturer's instructions. The first strand of cDNA reaction (1 μl) was subjected to RT–PCR amplification using gene-specific primers. The primers were designed using Vector NTI XI software (Invitrogen). The sequence of the primers is presented in Supplementary Table 2.

Quantification of mRNA was done with the SYBR Green method using ABI Prism 7300 Sequence Detection System (Applied Biosystems, Foster City, CA). Approximately 25 μl of reaction volume was used for the RT–PCR assay that consisted of (2 × 12.5 μl) Brilliant SYBR Green QPCR Master Mix (Applied Biosystems), 400 nM of primers (1 μl each from the stock), 11 μl of water and 0.5 μl of template. The thermal conditions consisted of an initial denaturation at 95 °C for 10 min followed by 40 cycles of denaturation at 95 °C for 15 s, annealing and extension at 60 °C for 1 min and a final step melting curve of 95 °C for 15 s, 60 °C for 15 s, and 95 °C for 15 s. All reactions were carried out in duplicate to reduce variation. The data were analyzed using SDS software version 2.0, and the results were exported to Microsoft Excel for further analysis. Data normalization was accomplished using the endogenous control β-actin and the normalized values were subjected to a $2^{-\Delta\Delta Ct}$ formula to calculate the fold change between the control and experimental groups.

**Fluorescence-activated cell sorting (FACS).** Satellite cells were analyzed by performing FACS analysis as described[38, 47]. Briefly cell suspensions prepared from 5-day-injured TA muscle of MyD88^{f/f} and MyD88^{myoKO} mice were immunostained with antibodies against CD45, CD31, Sca-1, and Ter-119 for negative selection (all PE conjugated, eBiosciences), and with α7-integrin (APC conjugated, Miltenyi Biotec) for positive selection. To detect CD45⁺ leukocytes, cell-suspensions prepared from 5-day-injured muscles of MyD88^{f/f} and MyD88^{myoKO} mice were incubated with PE-conjugated CD45 antibody (eBioscience) followed by FACS analysis. FACS analysis was performed on a C6 Accuri cytometer (BD Biosciences) equipped with two lasers.

**Statistical analyses and general experimental design.** The sample size was calculated using power analysis methods for a priori determination based on the s. d. and the effect size was previously obtained using the experimental procedures employed in the study. For animal studies, we estimated sample size from expected number of WT and MyD88-KO and MyD88^{myoKO} mice and their littermate MyD88^{f/f} controls. We calculated the minimal sample size for each group as eight animals. Considering a likely drop-off effect of 10%, we set sample size of each group of six mice. For some experiments, three to four animals were found sufficient to obtain statistical differences. Animals with same sex and same age were employed to minimize physiological variability and to reduce s.d. from mean. The exclusion criteria for animals were established in consultation with a veterinarian and experimental outcomes. In case of death, skin injury, ulceration, sickness, or weight loss of > 10%, the animal was excluded from analysis. Tissue samples were excluded in cases such as freeze artifacts on histological sections or failure in extraction of RNA or protein of suitable quality and quantity. We included animals from different breeding cages by random allocation to the different experimental groups. All animal experiments were blinded using number codes till the final data analyses were performed. Statistical tests were used as described in the Figure legends. Results are expressed as mean + s.d. Statistical analyses used two-tailed Student's t-test to compare quantitative data populations with normal distribution

and equal variance. A value of $p < 0.05$ was considered statistically significant unless otherwise specified.

**Data availability**. All relevant data related to this manuscript are available from the authors upon reasonable request.

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

## Acknowledgements

This work was supported by NIH grants AR068313, AR059810, and AG029623 to AK and AR069985 to SMH.

## Author contributions

S.M.H., J.S., R.P.F. and A.K. designed the work. S.M.H., J.S. and A.K. wrote the manuscript and all authors edited the manuscript. J.S., S.M.H., Y.S.G., A.R.S., A.S.-B., L.H. and G.X. performed the experiments.

## Additional information

**Competing interests:** The authors declare no competing financial interests.

