## [Peer Review File · Nature Communications]

Reviewers' comments:

Reviewer #1 (Remarks to the Author):

This study demonstrates the involvement of MyD88 in development and regeneration of skeletal muscle and shows that MyD88 promotes myoblast fusion through the activation of non-canonical NF- κ B signaling and canonical Wnt signaling pathways. The findings are novel; the experiments are well designed and utilize an impressive set of diverse experimental approaches. While the exact place of MYD88-mediated regulation among various steps of myogenesis that generate multinucleate myotubes remains to be clarified, the results can potential biomedical applications and, I expect, will be of importance to researchers working on different aspects of muscle biology.

Specific comments.

1) The authors report that MyD88 deficiency does not affect expression of markers of myogenic differentiation such as myogenin and MyHC (Fig. 2F) and suggest that MyD88 "is required specifically for myoblast fusion" and "promotes myoblast fusion through augmenting the expression of various profusion molecules". On the other hand, one of the major conclusions of the work is that MyD88 promotes myoblast fusion through the activation of canonical Wnt signaling pathway and this pathway has been reported to induce expression of different myogenic genes (for instance, Refs. 12, 14). How do the authors explain this apparent discrepancy?

2) The authors overstate their case for the importance of MyD88 in myoblast fusion and myogenesis by repeatedly using the word "required" as in "MyD88 is required specifically for myoblast fusion"; "MyD88 is required for myoblast fusion in vitro" and in "MyD88 is required for myoblast fusion during myogenesis". Their data show that MyD88-deficient mice have no major muscle phenotype and "there was no significant difference in muscle weight or myofiber CSA at the age of 3 months". While MyD88 deficiency inhibits fusion, myoblasts do fuse into myotubes with 5 to 10 nuclei (Figure 2). On a similar note, the phrase "MyD88 is essential for myoblast fusion during myogenesis" is an overstatement. Similarly, " MyD88 mediates myoblast fusion" suggests a direct (rather than regulatory) role for this protein in fusion.

3) Fig. 4. RD cells do not fuse and do not express MyHC. Forced expression of MyD88 partially rescues fusion. Is there any increase in MyHC expression?

4) Fig. 5. Please clarify the statistics. In the Methods I see "To measure average diameter of myotubes, 100-120 myotubes per group were included". In the figure legend it is "N=5 in each group.". Is N here (and in other figures) number of independent experiments?

5) Fig. 7, These experiments compare muscle regeneration for w.t and MyD88-deficient mice. How do you exclude possible effects of MyD88 deficiency on the post-injury inflammation and satellite cell activation? The scale bars are very small & difficult to find. This Figure should be re-organized to minimize the empty space.

6) Fig. 8 The authors conclude that overexpression of MyD88 promotes fusion of myoblasts with injured myofibers in vivo. How do you explain why there was no difference between numbers of mTmG+ myofibers for the muscles transplanted with vector alone myoblasts or with MyD88-overexpressing myoblasts? Can this lack of difference taken together with an increased size of mTmG+ myofibers indicate that to promote fusion MyD88 has to be expressed in both fusing cells? It can be interesting to study fusion between MyD88-expressing and MyD88-deficient myoblasts to

test whether expression of MyD88 in one of two fusing myoblasts is sufficient for efficient fusion.

7) " However, MyD88 promotes myoblast fusion only in the presence of the sequentially expressed conventional MRFs during myogenic differentiation. Overexpression of MyD88 does not promote fusion of non-myogenic cells or primary myoblasts incubated in growth medium." Where are the data supporting these statements?

8) What are "transmembrane lipids" (p.3)? Please correct.

Reviewer #2 (Remarks to the Author):

In this manuscript, the authors investigate the myogenic role for MyD88, a key adaptor protein for interleukin-1 and toll-like receptors that has a well-known function in the innate immune response. Recruitment of MyD88 facilitates activation of various transcription factors, such as NF- κ B, leading to induction of various inflammatory cytokines. NF- κ B signaling has been implicated in myoblast fusion, though the role of MyD88 in myogenesis is unclear. Shin et al., employed loss of function and gain of function strategies to assess the role of MyD88 during in vivo and in vitro myogenesis, and reported an important contribution of MyD88 in myoblast fusion. MyD88 is shown to regulate postnatal skeletal muscle growth, myotube formation in vitro, and regeneration following injury and load-induced hypertrophy in adult muscles. The authors also identified non-canonical NF- κ B and canonical Wnt signaling and activation of various profusion molecules as mechanisms through which MyD88 augments myogenesis. While these findings extend our knowledge of the fusion mechanisms of skeletal muscle growth and development, enthusiasm is somewhat tempered by an overall lack of a robust phenotype in vivo and over-interpretations of key findings. Critiques of the manuscript and suggestions are listed below:

1. In general, the LOF and GOF phenotypes observed in vivo and in vitro are not visually robust compared to respective controls, especially in Figures 1D & E, 3A, 5D, 7A, 9B, and S5C. To be fair, the low resolution of images presented (the H & E stains and in vitro myosin images) may mask some observable differences in phenotypes. We suggest using immunostaining with an antibody that outlines the myofiber (dystrophin, laminin) for in vivo sections. Overall, the authors should provide images more representative of their quantitative data.
2. The authors state there are more mono-nucleated cells in the MyD88 KO sample in Figure 1D. They are stating this to provide evidence for lack of fusion, however this is an example of over-interpretation. First, from the images I am unable to see an increase in mono-nuclear cells. Secondly, the lineage of these mono-nuclear cells is not reported. Their interpretation would be more appropriate if they stained for myh3 (embryonic myosin) or myogenin and clearly showed these were mono-nucleated. Moreover, this observation could be due to loss of MyD88 in non-muscle cells.
3. Another example of over-interpretation relates to the RMS experiment. MyD88 ONLY weakly increases fusion after the cells are induced to become myogenic (through expression of a constitutively active MKK6), but essentially has no effect in cultures not myogenically activated. And the repression of MyD88 in RMS cells is not surprising given that the majority of differentiation genes are reduced (they even show a reduction in myosin). I think using this sub-optimal data to even suggest that this could be a valuable therapeutic for RMS is grossly misleading. I would even argue that this piece of data could be excluded as it does not add much to their story. Lastly, with regards to fusion, while data in the current study do support a role for MyD88 in mediating myoblast fusion in vitro, it is an overstatement to claim that MyD88 is required or essential in this process as described throughout the manuscript.
4. Is there a reduction in CSA in MyD88 mKO mice after barium chloride injury? Additionally, the authors should quantify the number and percentage of regenerating myofibers to ascertain if regeneration is compromised following injury with MyD88 deletion in myoblasts. As MyD88 is essential for innate immunity, it is curious that the authors do not observe a more severe phenotype (e.g. excess infiltration of inflammatory cells, increased swelling) in their global MyD88-KO mice following injury (Fig. S5C). Could the authors explain this discrepancy? Furthermore, the

authors should clarify if mice were treated with BaCl₂ at 8-12 weeks old (as stated in experimental procedures), or at 3-mo old (as stated in results and figure legends). Mice at 8 weeks old may not fully developed and data generated at this age (especially myofiber CSA) should not be included in the analyses of adult skeletal muscle regeneration.

5. The authors mention that MyD88 mKO mice do not exhibit reduced CSA at 3 months of age, however they do comment further. Does this suggest that the kinetics of fusion are just slower in MyD88 mKO mice and they eventually catch up? Or do the 3 month old MyD88 mKO myofibers exhibit reduced number of myonuclei, but this number is enough to support growth? The authors could get at these questions by isolating single myofibers and quantifying nuclei/myofiber. Or maybe this suggests that MyD88 does not directly augment fusion.

6. Overall, the discussion is simply a rehash of the results and does not deal with the important issues. For instance, our comment #5 above was not mentioned. Additionally, why is there such a modest fusion phenotype in the MyD88-KO, especially compared to their in vitro data? Also why there is no CSA phenotype in MyD88 KO after injury (Figure 7)? Does a more pronounced muscle phenotype in MyD88-KO compared to MyD88 mKO suggest that MyD88 is also functioning in non-muscle cells to augment fusion? I realize these are difficult questions that may not indicate a direct role for MyD88 during myoblast fusion, but they should still be discussed.

7. To substantiate the claim that "MyD88 promotes skeletal muscle growth in adult mice through augmenting myoblast fusion" (results), the authors need to directly assess and quantify the degree of fusion in their control and myoblast-specific KO mouse following overload.

8. The authors clearly show that fusion, though diminished, does occur in the absence of MyD88 (Figure 2) through immunostaining and quantification of smaller myotubes with fewer nuclei. To strengthen overall findings, it would be informative to reveal the total number of myotubes for existing samples. This would provide insight on the role of MyD88 in primary fusion (fusion of myoblasts to generate nascent myotubes) compared to secondary fusion (myonuclear addition in existing myotubes) and add to understanding of MyD88 function.

9. In Figure 5, the authors investigate the non-canonical NF- κ B pathway in myoblasts and show that in MyD88 KO cells there is a reduction of p52 cleavage. Overall the authors think that MyD88 promotes p52 cleavage. They then rescue MyD88 KO myoblasts with expression of p52, RelB, IKK α . How this rescues fusion, even in the absence of MyD88, is not clear. If the normal function for MyD88 in myoblasts is to promote cleavage of p52, then how does adding more p52 overcome the effect of loss of MyD88? Does something else cleave p52 in the absence of MyD88 but only when p52 is extremely high? The authors show an increase in p52 in 5G but do not show cleavage. The authors should deal with these issues.

Minor points:

1. Muscle weights are shown from MyD88 KO mice in Figure 1, demonstrating reduced muscle mass. However, these weights are not normalized. Additionally, it is not reported whether non-muscle tissues are also smaller? If so, it could suggest that MyD88 regulates tissue size through fusion-independent mechanisms.

2. Figure 1F was not referred to in the text.

3. The authors should provide a reference for the use of LDH as a marker for cell death. They mention it has been extensively used on page 8 but no reference is listed.

4. In Figure 8, though the authors used mTmG mice for Myd88 overexpression experiments, they assessed and quantified tomato+ myofibers. Hence, headings in Figures 8B and 8C should be changed to reflect that.

5. In Figure 9 legends, * was used to denote differences between corresponding shams, as well as between 14d-loaded muscle of control mice; however # should be used to depict differences for the latter finding, as shown in Figures 9C and 9D.

6. On page 10 of results, the authors mentioned increased mRNA levels of M-cadherin and N-cadherin, even though they are not statistically significant, but omitted Adam-12 even though that molecule demonstrated the most robust change in expression following MyD88. This should be revised accordingly.

7. The authors do not report the percentage of transfected cells in any of their experiments. This is an important factor for proper interpretation and could even account for some less than robust effects if they are dealing with heterogenous cultures.

8. The heading "MyD88 is required for load-induced myofiber hypertrophy" is not completely accurate (Fig. 9) as ~20% increase in myofiber CSA was observed in MyD88 mKO plantaris 14 d after surgery. The heading used in the results section "MyD88 mediates overload-induced muscle hypertrophy in adult mice" is therefore more appropriate.

Reviewer #3 (Remarks to the Author):

1. The authors emphasize as an important conclusion of their work that MyD88 acts in a receptor and cell autonomous manner in promoting myoblast fusion. However, the data in support of this is rather limited. They show that myoblast fusion is not promoted by agonist to a few of the receptors known to utilize MyD88, including TLR4, TLR3, TLR2, and TLR7. However, several other TLRs and IL-1 all lead to myosome formation in signaling. The authors should rule out the roles of other TLRs and IL-1 receptor signaling in the responses.

2. Along the same lines, does MyD88 form aggregates as seen with receptor mediated activation involving MyD88 and where does MyD88 localize in the cell?

3. The investigators provide supportive data that both non-canonical NFKB and Wnt3a expression/signaling are downstream of MyD88 but did not link these 2 pathways. Does blocking non-canonical NFKB signaling block Wnt3a expression? Addressing, this question does not seem to be out of the scope of the studies.

4. The investigators should deal with published results showing that MyD88 is involved in skeletal muscle injury in a model of hind limb ischemia and that deletion MyD88 appears to be associated with a higher skeletal muscle nuclear content and myofibril size after skeletal muscle injury (Sachdev, et al, PLOS One, 2012 and Physiol. Rep., 2014). These contrasting results emphasize the important of MyD88 in regulating numerous processes relative to skeletal muscle function and regeneration after injury.

5. The authors state they wanted to assess how MyD88 was upregulated in myoblasts but, in fact, have done little to provide insight. For example, did DM induce an increase in MyD88 mRNA levels, MyD88 transcription, or MyD88 translation or stabilization?

6. Is MyD88 increased in RMS? This is important as the authors emphasize the RMS as a clinically relevant scenario involving the apparent activation of this process.

7. Did the Myod1-cre mice show any abnormality in skeletal muscle?

8. Figure 9 presents a critical and exciting finding that in skeletal muscle hypertrophy (independent of skeletal muscle direct injury) leads to increased MyD88 expression and a role for MyD88 in muscle hypertrophy. However, the numbers of mice (n=3) is rather small. In an era of an emphasis on rigor and reproducibility in science, this small n number raises concerns.

RESPONSE to REVIEWERS' COMMENTS

We sincerely thank all the three reviewers for their positive comments about the importance of and novelty of our work and providing highly constructive suggestions. We have now performed several new experiments to address the reviewers' comments. Thanks to reviewers' suggestions, we believe our manuscript has now been considerably improved. All the major changes in the manuscript are highlighted by yellow background. Our point-wise response to reviewers' comments is as follows:

REVIEWER #1

This study demonstrates the involvement of MyD88 in development and regeneration of skeletal muscle and shows that MyD88 promotes myoblast fusion through the activation of non-canonical NF- κ B signaling and canonical Wnt signaling pathways. The findings are novel; the experiments are well designed and utilize an impressive set of diverse experimental approaches. While the exact place of MYD88-mediated regulation among various steps of myogenesis that generate multinucleate myotubes remains to be clarified, the results can potential biomedical applications and, I expect, will be of importance to researchers working on different aspects of muscle biology.

OUR RESPONSE: We sincerely thank the reviewer for all these positive comments. Specific comments.

1) The authors report that MyD88 deficiency does not affect expression of markers of myogenic differentiation such as myogenin and MyHC (Fig. 2F) and suggest that MyD88 “is required specifically for myoblast fusion” and “promotes myoblast fusion through augmenting the expression of various profusion molecules”. On the other hand, one of the major conclusions of the work is that MyD88 promotes myoblast fusion through the activation of canonical Wnt signaling pathway and this pathway has been reported to induce expression of different myogenic genes (for instance, Refs. 12, 14). How do the authors explain this apparent discrepancy?

OUR RESPONSE: As the reviewer knows, Wnt signaling is quite complex, which involves 19 Wnt ligands, as well as several receptors and co-receptors. Both canonical and non-canonical pathways regulate multiple aspects of myogenesis. During embryonic development, Wnt signaling (attributed to increased levels of Wnt1, Wnt7a or Wnt6) has been shown to influence the expression of MRFs especially Myf5 and MyoD (Trends Cell Biol. 2012 Nov;22(11):602-9). However, the role of canonical Wnt signaling in adult skeletal muscle regeneration and during *in vitro* myogenesis is relatively less described. In early phase of muscle regeneration, the expression of three Wnt ligands: Wnt5a, Wnt5b, and Wnt7a is increased, whereas the levels of Wnt4 are decreased. At later stages of regeneration, levels of Wnt7b and Wnt3a are induced. Wnt7a, which functions through the PCP pathway, is required to promote satellite cell expansion during regenerative myogenesis. By contrast, canonical signaling is required for the differentiation of myogenic cells (reviewed in Trends Cell Biol. 2012 Nov;22(11):602-9). Cell culture studies have shown that stimulation of canonical Wnt signaling leads to the formation of larger myotubes which may be attributed to both increased myogenic differentiation and myoblast fusion (Mol Biol Cell. 2004 Oct;15(10):4544-55). Indeed, further detailed investigations have found a segregated regulation of myoblast fusion and muscle-specific gene

expression in response to the activation of Wnt signaling following stimulation of myogenic differentiation. For example, it has been reported that while both Wnt3a and LiCl (an activator of canonical Wnt signaling) promote myoblast fusion, muscle-specific gene expression was increased by LiCl, but not by Wnt-3a or β -catenin over-expression (Pansters et al. Cell Mol Life Sci. 2011 Feb;68(3):523-35). A more recent study has dissected the role of Wnt/ β -catenin signaling in regulation of various steps of myogenesis. The study shows that Wnt/ β -catenin signaling specifically regulates myoblast fusion during *in vitro* differentiation, although authors did not measure the markers of muscle differentiation (Suzuki et al., Mol Cell Biol. 2015 May;35(10):1763-76). We would also like to emphasize that previous studies have not thoroughly investigated the effect of inhibition of Wnt/ β -catenin signaling on the expression of myogenic markers during adult muscle regeneration and *in vitro* differentiation. As a matter of fact, we were able to find only two studies that utilized various forms of canonical Wnt inhibitors in the context of muscle regeneration and *in vitro* muscle formation. The first study by Brack et al (Cell Stem Cell. 2008 Jan 10;2(1):50-9) showed that inhibition of Wnt signaling via sFRP3 treatment had no effect on muscle regeneration when administered at early stage of muscle regeneration (i.e. 2 days), while treatment at a later stage (i.e. 3.5 days post injury) impaired formation of new muscle fibers; however the expression of muscle differentiation markers was not investigated. In addition, sFRP3, which was utilized in this report, functions by antagonizing Wnt8 signaling specifically, as opposed to Wnt3a which was investigated in our manuscript. Another study employing small molecule inhibitors found that FH535 (which interferes with β -catenin/Tcf complex formation) and GW9662 (which is structurally similar to FH535, but does not interfere with complex formation) had no effect on Troponin T (marker of myogenic differentiation) expression in differentiating C2C12 myoblast cultures (Tanka et al. J Mol Signal. 2011 Oct 5;6:12). Finally, a report by Tran et al (J Biol Chem. 2012 Sep 21;287(39):32651-64) had showed that heparan sulfate 6-O-endosulfatases (Sulfs) promote muscle regeneration through canonical Wnt mediated regulation of myoblast fusion. Their findings demonstrate that while the size of eMyHC⁺ fibers and centrally localized nuclei are significantly reduced in regenerating muscle of skeletal muscle-specific Sulf1/2-double null mice, the number of eMyHC⁺ is increased, indicating that Sulf deletion specifically impaired myoblast fusion and not myogenic differentiation through blunted Wnt signaling. In our study, we found that the deletion of MyD88 does not affect the expression of MyHC but it drastically reduces formation of multinucleated myotubes. We believe that this is partially due to decreased canonical Wnt signaling as activating this pathway with Wnt3a or LiCl treatment rescued the fusion defect in MyD88-KO cultures. We would also like to bring to the reviewers' attention that deletion of MyD88 in myoblasts reduces but does not completely block Wnt signaling, which may also explain why the expression of myogenic markers is not affected. In summary, Wnt signaling is a highly complex process and different components of this pathway may be involved in regulation of specific steps of myogenesis including but not limited to myogenic differentiation and fusion. Moreover, Wnt signaling may also be involve in signaling cross-talk with other pathways that may in turn augment myogenesis.

2) The authors overstate their case for the importance of MyD88 in myoblast fusion and myogenesis by repeatedly using the word "required" as in "MyD88 is required specifically for myoblast fusion"; "MyD88 is required for myoblast fusion *in vitro*" and in "MyD88 is required for myoblast fusion during myogenesis". Their data show that MyD88-deficient mice have no major muscle phenotype and "there was no significant difference in muscle weight or myofiber CSA at the age of 3 months". While MyD88 deficiency inhibits fusion, myoblasts do fuse into

myotubes with 5 to 10 nuclei (Figure 2). On a similar note, the phrase “MyD88 is essential for myoblast fusion during myogenesis” is an overstatement. Similarly, “MyD88 mediates myoblast fusion” suggests a direct (rather than regulatory) role for this protein in fusion.

OUR RESPONSE: We completely agree with the reviewer that our statements were quite strong in the previous submission. We agree that words such as “Regulates” “promotes” or “mediates” are more appropriate in this context. We have now changed these statements throughout the manuscript.

3) Fig. 4. RD cells do not fuse and do not express MyHC. Forced expression of MyD88 partially rescues fusion. Is there any increase in MyHC expression?

OUR RESPONSE: The overexpression of MyD88 does not augment the expression of MyHC. It only increase the fusion when a constitutively active mutant of MKK6 (upstream of p38MAPK), which promotes differentiation of RD cells, was overexpressed. Our RD cells experiments only suggested that MyD88 is up-regulated and mediates myoblast fusion only in the presence of an intact myogenic program which is also now evident by results presented in Supplementary Fig. 6. The second reviewer had suggested that these experiments with RD cells do not contribute much to the manuscript. Therefore we have now excluded all experiments with RD cells from our revised manuscript.

4) Fig. 5. Please clarify the statistics. In the Methods I see “To measure average diameter of myotubes, 100-120 myotubes per group were included”. In the figure legend it is “N=5 in each group.”. Is N here (and in other figures) number of independent experiments?

OUR RESPONSE: Yes. N defines the number of independent experiment. We have now stated this in the “Experimental Procedures” section of our revised manuscript (Page 32).

5) Fig. 7, these experiments compare muscle regeneration for w.t and MyD88-deficient mice. How do you exclude possible effects of MyD88 deficiency on the post-injury inflammation and satellite cell activation? The scale bars are very small & difficult to find. This Figure should be re-organized to minimize the empty space.

OUR RESPONSE: We have now provided a larger scale bar (now Fig. 6D). We have also now studied myofiber regeneration, satellite cell activation, and inflammatory response at day 5 post-injury. Our results demonstrate that myofiber CSA is significantly reduced in regenerating TA muscle of MyD88^{myoKO} mice compared with littermate control mice (Fig. 6A-C). By performing FACS analysis, we have also quantified satellite cells and inflammatory immune cells in injured TA muscle of these mice. We found no significant difference in the proportion of satellite cells ($\alpha7$ -Intergin⁺) or inflammatory cells (CD45⁺ cells) in injured TA muscle of MyD88^{f/f} and MyD88^{myoKO} (Please refer to Supplementary Fig. 10) suggesting that the myoblast-specific deletion of MyD88 does not affect satellite cell content or inflammatory response during muscle regeneration.

6) Fig. 8. The authors conclude that overexpression of MyD88 promotes fusion of myoblasts with injured myofibers in vivo. How do you explain why there was no difference between numbers of mTmG⁺ myofibers for the muscles transplanted with vector alone myoblasts or with MyD88-overexpressing myoblasts? Can this lack of difference taken together with an increased size of mTmG⁺ myofibers indicate that to promote fusion MyD88 has to be expressed in both fusing cells? It can be interesting to study fusion between MyD88-expressing and MyD88-

deficient myoblasts to test whether expression of MyD88 in one of two fusing myoblasts is sufficient for efficient fusion.

OUR RESPONSE: We would like to clarify that for *in vivo* experiment (now Fig. 7) we have used wild-type mice. Furthermore, the transplanted mT⁺ myoblasts are also WT but are overexpressing MyD88 cDNA therefore both fusion partners in this case are expressing MyD88 yet at different levels. There is no statistically significant difference, however, the trend is towards a reduced number of mT⁺ myofibers. We believe this could be attributed to the fact that MyD88-overexpressing myoblasts have enhanced capacity to fuse with injured myofibers and hence more myoblasts can fuse with the same myofiber leading to increased size of individual myofibers but reduced overall number of engrafted fibers. To further ascertain this is the case, we have now also measured the fluorescence intensity of the mT⁺ engrafted myofibers. Our new results demonstrate that mean fluorescence intensity is significantly higher in the mT⁺ myofibers originating from MyD88-overexpressing mT⁺ myoblasts compared to those transfected with empty vector only (Fig. 7D). However, we agree with the reviewer that it is important to understand whether MyD88 is needed in both fusion partners. We have now performed cell mixing experiments to address this issue. Our results show that MyD88-KO myoblasts are able to fuse with WT myoblasts, albeit with highly reduced capacity (Supplementary Fig. 4).

7) “However, MyD88 promotes myoblast fusion only in the presence of the sequentially expressed conventional MRFs during myogenic differentiation. Overexpression of MyD88 does not promote fusion of non-myogenic cells or primary myoblasts incubated in growth medium.” Where are the data supporting these statements?

OUR RESPONSE: We have now included these results in the Supplemental data file. Please refer to Supplementary Fig. 6.

8) What are “transmembrane lipids” (p.3)? Please correct.

OUR RESPONSE: This was transmembrane protein. We have now corrected this in our manuscript.

REVIEWER #2

In this manuscript, the authors investigate the myogenic role for MyD88, a key adaptor protein for interleukin-1 and toll-like receptors that has a well-known function in the innate immune response. Recruitment of MyD88 facilitates activation of various transcription factors, such as NF- κ B, leading to induction of various inflammatory cytokines. NF- κ B signaling has been implicated in myoblast fusion, though the role of MyD88 in myogenesis is unclear. Shin et al., employed loss of function and gain of function strategies to assess the role of MyD88 during *in vivo* and *in vitro* myogenesis, and reported an important contribution of MyD88 in myoblast fusion. MyD88 is shown to regulate postnatal skeletal muscle growth, myotube formation *in vitro*, and regeneration following injury and load-induced hypertrophy in adult muscles. The authors also identified non-canonical NF- κ B and canonical Wnt signaling and activation of various profusion molecules as mechanisms through which MyD88 augments myogenesis. While these findings extend our knowledge of the fusion mechanisms of skeletal muscle growth and development, enthusiasm is somewhat tempered by an overall lack of a robust phenotype *in vivo* and over-interpretations of key findings. Critiques of the manuscript and suggestions are listed below:

1. In general, the LOF and GOF phenotypes observed *in vivo* and *in vitro* are not visually robust compared to respective controls, especially in Figures 1D & E, 3A, 5D, 7A, 9B, and S5C. To be fair, the low resolution of images presented (the H & E stains and *in vitro* myosin images) may mask some observable differences in phenotypes. We suggest using immunostaining with an antibody that outlines the myofiber (dystrophin, laminin) for *in vivo* sections. Overall, the authors should provide images more representative of their quantitative data.

OUR RESPONSE: We agree with the reviewers that some of our images especially those stained with H&E were not of good quality. However, we would also like to mention the quality of images is considerably diminished when converting them into PDF and then inserting into the main manuscript and again making PDF of the whole manuscript. Our original images look much sharper and are of publication quality. Per reviewer's suggestion, we have now performed dystrophin or laminin staining on the muscle sections for all our *in vivo* studies. We hope reviewer will find new images of improved quality and the phenotype is now clearly visible.

2. The authors state there are more mono-nucleated cells in the MyD88 KO sample in Figure 1D. They are stating this to provide evidence for lack of fusion, however this is an example of over-interpretation. First, from the images I am unable to see an increase in mono-nuclear cells. Secondly, the lineage of these mono-nuclear cells is not reported. Their interpretation would be more appropriate if they stained for myh3 (embryonic myosin) or myogenin and clearly showed these were mono-nucleated. Moreover, this observation could be due to loss of MyD88 in non-muscle cells.

OUR RESPONSE: We agree with the reviewer that our images were not of good quality and our statement about more mononucleated cells may not be correct due to lack of specific staining. We have now performed immunostaining for eMyHC, MyoD, and Pax7 on muscle sections. There were very few eMyHC⁺ fibers (3-4 per section) and there was no difference between WT and MyD-KO mice. We also did not find any MyoD⁺ cells. By contrast, we found that there was a small increase (but not statistically significant, $p=0.061$) in the number of Pax7⁺ satellite cells in skeletal muscle of 2-week old MyD88^{myoKO} mice compared to corresponding control mice (Supplementary Fig. 1B and 1C).

3. Another example of over-interpretation relates to the RMS experiment. MyD88 ONLY weakly increases fusion after the cells are induced to become myogenic (through expression of a constitutively active MKK6), but essentially has no effect in cultures not myogenically activated. And the repression of MyD88 in RMS cells is not surprising given that the majority of differentiation genes are reduced (they even show a reduction in myosin). I think using this sub-optimal data to even suggest that this could be a valuable therapeutic for RMS is grossly misleading. I would even argue that this piece of data could be excluded as it does not add much to their story. Lastly, with regards to fusion, while data in the current study do support a role for MyD88 in mediating myoblast fusion *in vitro*, it is an overstatement to claim that MyD88 is required or essential in this process as described throughout the manuscript.

OUR RESPONSE: We agree with the reviewer that RMS cell results were just supportive and did not add much to our manuscript. Per reviewer's suggestion we have now excluded RMS data from our manuscript. As mentioned above in response to similar comments by Reviewer #1, we have now toned down our sentences. We now state that MyD88 mediates myoblast fusion instead of claiming it is essential for myoblast fusion.

4. Is there a reduction in CSA in MyD88 mKO mice after barium chloride injury? Additionally, the authors should quantify the number and percentage of regenerating myofibers to ascertain if regeneration is compromised following injury with MyD88 deletion in myoblasts. As MyD88 is essential for innate immunity, it is curious that the authors do not observe a more severe phenotype (e.g. excess infiltration of inflammatory cells, increased swelling) in their global MyD88-KO mice following injury (Fig. S5C). Could the authors explain this discrepancy? Furthermore, the authors should clarify if mice were treated with BaCl₂ at 8-12 weeks old (as stated in experimental procedures), or at 3-mo old (as stated in results and figure legends). Mice at 8 weeks old may not fully developed and data generated at this age (especially myofiber CSA) should not be included in the analyses of adult skeletal muscle regeneration.

OUR RESPONSE: We have now performed muscle injury experiment in MyD88^{myoKO} mice and performed eMyHC staining and measured CSA. We observed a significant decrease in CSA of eMyHC⁺ myofibers in MyD88^{myoKO} mice compared to controls. Please refer to new Figures 6A-C. We agree with the reviewer that MyD88 is involved in the inflammatory response in some pathological conditions. However, with the exception of reduced myofiber CSA in regenerating skeletal muscle of whole body MyD88-KO mice (now Supplementary Fig. 9), we did not find any major difference including cellular infiltrates etc. (now Supplementary Fig. 9). Since the focus of the paper is on the role of MyD88 in myoblasts, we have now specifically investigated the activation of satellite cells and inflammatory immune response in regenerating skeletal muscle of MyD88^{ff} and MyD88^{myoKO} mice. Our results demonstrated that the deficiency of MyD88 in myoblasts did not affect the number of satellite cells or immune cells (i.e. CD45⁺) in regenerating muscle of MyD88^{myoKO} mice (Supplementary Fig. 10).

The 8-12 week was just a typo in the manuscript. We have used 12-week old littermate mice for all of our experiments. This has been corrected in the manuscript (please refer to highlighted text on Page # 8 and 21).

5. The authors mention that MyD88 mKO mice do not exhibit reduced CSA at 3 months of age, however they do comment further. Does this suggest that the kinetics of fusion are just slower in MyD88 mKO mice and they eventually catch up? Or do the 3 month old MyD88 mKO myofibers exhibit reduced number of myonuclei, but this number is enough to support growth? The authors could get at these questions by isolating single myofibers and quantifying nuclei/myofiber. Or maybe this suggests that MyD88 does not directly augment fusion.

OUR RESPONSE: This is an excellent suggestion by the reviewer. We have now performed this experiment using isolated single myofibers from 3-month old MyD88^{ff} and MyD88^{myoKO} mice. Our results demonstrate that the overall number of myonuclei per myofiber is indeed significantly reduced in MyD88^{myoKO} compared to control mice yet is sufficient to support muscle growth to the level of wild type counterparts (Please refer to Supplementary Fig. 2). We have now included and discussed these results in this revised submission. .

6. Overall, the discussion is simply a rehash of the results and does not deal with the important issues. For instance, our comment #5 above was not mentioned. Additionally, why is there such a modest fusion phenotype in the MyD88-KO, especially compared to their in vitro data? Also why there is no CSA phenotype in MyD88 KO after injury (Figure 7)? Does a more pronounced muscle phenotype in MyD88-KO compared to MyD88 mKO suggest that MyD88 is also functioning in non-muscle cells to augment fusion? I realize these are difficult questions that

may not indicate a direct role for MyD88 during myoblast fusion, but they should still be discussed.

OUR RESPONSE: We have now considerably improved the discussion part of our manuscript. We have also included the points suggested by the reviewer. In the last submission, we did not present data about myofiber CSA of regenerating muscle of MyD88^{fl/fl} and MyD88^{myoKO} mice. We have now performed muscle injury experiment with MyD88^{myoKO} mice similar to what was done with MyD88-KO mice. Our experiment demonstrates that there is significant reduction in myofiber CSA in MyD88^{myoKO} mice at 5 days post injury (now Fig. 6C) which is comparable to what we observed in MyD88-KO mice (in Supplementary Fig. 9E). We have also provided a possible explanation as to why the phenotype is modest *in vivo* compared to *in vitro* in the “Discussion” section of this revised manuscript (highlighted text on page # 26).

7. To substantiate the claim that “MyD88 promotes skeletal muscle growth in adult mice through augmenting myoblast fusion” (results), the authors need to directly assess and quantify the degree of fusion in their control and myoblast-specific KO mouse following overload.

OUR RESPONSE. This is an excellent suggestion. We have now specifically assessed the degree of fusion by labelling myoblasts with EdU in the synergistic ablation experiment. We found a significant reduction in the number of EdU⁺ myonuclei in overloaded muscle of MyD88^{myoKO} mice compared to corresponding control mice. Please refer to new Figures 8E and 8F in our revised experiment.

8. The authors clearly show that fusion, though diminished, does occur in the absence of MyD88 (Figure 2) through immunostaining and quantification of smaller myotubes with fewer nuclei. To strengthen overall findings, it would be informative to reveal the total number of myotubes for existing samples. This would provide insight on the role of MyD88 in primary fusion (fusion of myoblasts to generate nascent myotubes) compared to secondary fusion (myonuclear addition in existing myotubes) and add to understanding of MyD88 function.

OUR RESPONSE: We have now counted the number of myotubes. Please refer to Fig. 2F and 2G in our revised manuscript. We did not find any difference in the overall number of myotubes containing 2 or more nuclei between WT and MyD88-KO cultures indicating that most MyD88 deficient myoblasts are capable of undergoing primary fusion and forming nascent myotubes. However, we also observed that not all MyD88-KO myoblast underwent primary fusion and a significant number although expressed MyHC, remained mononucleated. Based on this analysis, we believe that MyD88 primarily mediates secondary fusion and is involved but to a lesser extent in mediating primary fusion. This has now been discussed in the revised manuscript (highlighted text on Page # 9 and # 22).

9. In Figure 5, the authors investigate the non-canonical NF- κ B pathway in myoblasts and show that in MyD88 KO cells there is a reduction of p52 cleavage. Overall the authors think that MyD88 promotes p52 cleavage. They then rescue MyD88 KO myoblasts with expression of p52, RelB, IKK α . How this rescues fusion, even in the absence of MyD88, is not clear. If the normal function for MyD88 in myoblasts is to promote cleavage of p52, then how does adding more p52 overcome the effect of loss of MyD88? Does something else cleave p52 in the absence of MyD88 but only when p52 is extremely high? The authors show an increase in p52 in 5G but do not show cleavage. The authors should deal with these issues.

OUR RESPONSE: Upon activation of non-canonical NF- κ B signaling pathway, IKK α phosphorylates p100 protein. Upon phosphorylation, p100 undergoes partial proteolytic degradation leading to the formation of p52 subunit. The p52 subunit makes a dimer with RelB through which both molecules translocate to the nucleus and bind to DNA to increase gene expression. The proteolytic processing of p100 into p52 protein is a critical step in the activation of non-canonical NF- κ B signaling. We used cDNA which express only the p52 part of the p100 subunit. In this way, the proteolytic processing step is bypassed because p52 is the only functional part of p100/p52 NF- κ B subunit. Overexpression of p52 protein is a common approach to activate non-canonical NF- κ B pathway independent of upstream signaling. Since we overexpressed p52 cDNA we showed only p52 protein levels in Fig. 4F. Overexpression of p52 does not affect the levels of p100 protein which is also evident in the uncropped gel image included in the Supplementary Fig. 11 of this manuscript.

Minor points:

1. Muscle weights are shown from MyD88 KO mice in Figure 1, demonstrating reduced muscle mass. However, these weights are not normalized. Additionally, it is not reported whether non-muscle tissues are also smaller? If so, it could suggest that MyD88 regulates tissue size through fusion-independent mechanisms.

OUR RESPONSE: We have now presented muscle weight normalized to whole body weight. We had also collected spleen and kidney from these mice. As our results in Fig. 1C demonstrate, there is a significant reduction in weight in most muscle groups when normalized to total body weight, but not in kidney or spleen, suggesting that MyD88 specifically affects skeletal muscle.

2. Figure 1F was not referred to in the text.

OUR RESPONSE: This has now been referred (on Page # 6).

3. The authors should provide a reference for the use of LDH as a marker for cell death. They mention it has been extensively used on page 8 but no reference is listed.

OUR RESPONSE: We have now cited three references for this (ref. 43-45).

4. In Figure 8, though the authors used mTmG mice for Myd88 overexpression experiments, they assessed and quantified tomato⁺ myofibers. Hence, headings in Figures 8B and 8C should be changed to reflect that.

OUR RESPONSE: The reviewer is correct. We have now corrected this to mT⁺ (now Figure 7).

5. In Figure 9 legends, * was used to denote differences between corresponding shams, as well as between 14d-loaded muscle of control mice; however # should be used to depict differences for the latter finding, as shown in Figures 9C and 9D.

OUR RESPONSE: This was an oversight on our part. It has now been corrected (now Fig. 8C and 8D).

6. On page 10 of results, the authors mentioned increased mRNA levels of M-cadherin and N-cadherin, even though they are not statistically significant, but omitted Adam-12 even though that molecule demonstrated the most robust change in expression following MyD88. This should be revised accordingly.

OUR RESPONSE: We agree with the reviewer. This was again an oversight on our part. We have now mentioned this in the “Results” section (on Page # 11, 12).

7. The authors do not report the percentage of transfected cells in any of their experiments. This is an important factor for proper interpretation and could even account for some less than robust effects if they are dealing with heterogeneous cultures.

OUR RESPONSE: We used the electroporation approach for overexpression studies. We consistently get ~80% cells transfected. We have now mentioned this in the “Experimental procedure” section (Page # 31).

8. The heading “MyD88 is required for load-induced myofiber hypertrophy” is not completely accurate (Fig. 9) as ~20% increase in myofiber CSA was observed in MyD88 mKO plantaris 14 d after surgery. The heading used in the results section “MyD88 mediates overload-induced muscle hypertrophy in adult mice” is therefore more appropriate.

OUR RESPONSE: We agree with the reviewer. We have revised the heading accordingly (Page # 19).

REVIEWER #3

1. The authors emphasize as an important conclusion of their work that MyD88 acts in a receptor and cell autonomous manner in promoting myoblast fusion. However, the data in support of this is rather limited. They show that myoblast fusion is not promoted by agonist to a few of the receptors known to utilize MyD88, including TLR4, TLR3, TLR2, and TLR7. However, several other TLRs and IL-1 all lead to myosome formation in signaling. The authors should rule out the roles of other TLRs and IL-1 receptor signaling in the responses.

OUR RESPONSE: We have now investigated the role of IL-1 β and agonists of TLR1 to TLR9. However, we did not find any increase in myoblast fusion after treatment with TLR agonists or IL-1 β . The data are now presented in Supplementary Fig. 7A.

2. Along the same lines, does MyD88 form aggregates as seen with receptor mediated activation involving MyD88 and where does MyD88 localize in the cell?

OUR RESPONSE: We have now performed immunostaining of myoblasts and myotubes with MyD88 antibody. We found that MyD88 is localized in the cytoplasm. While we see an increase in MyD88 levels by immunostaining after incubation of myoblasts in differentiation medium, there was no evidence of MyD88 aggregate formation. As a positive control, we treated myoblasts with LPS and observed an increase in aggregate formation. The results are presented in Supplementary Fig. 7B. These results further indicate that MyD88 regulates myoblast fusion in a cell-autonomous manner.

3. The investigators provide supportive data that both non-canonical NF-KB and Wint3a expression/signaling are downstream of MyD88 but did not link these 2 pathways. Does blocking non-canonical NF-KB signaling block Wint3a expression? Addressing, this question does not seem to be out of the scope of the studies.

OUR RESPONSE: This is a really good suggestion. We have now performed an experiment to provide this link. Interestingly, we found that siRNA-mediated knockdown of p100/52 reduced

the levels of Wnt3a in differentiating myoblasts. Similarly, we found that the levels of GSK-3 β phosphorylation (at ser9) were also reduced upon knockdown of p100/p52. These results suggest that non-canonical NF- κ B signaling activates Wnt3a expression and canonical Wnt signaling during myogenesis. Please refer to new Figure 5H for these results.

4. The investigators should deal with published results showing that MyD88 is involved in skeletal muscle injury in a model of hind limb ischemia and that deletion MyD88 appears to be associated with a higher skeletal muscle nuclear content and myofibril size after skeletal muscle injury (Sachdev, et al, PLOS One, 2012 and Physiol. Rep., 2014). These contrasting results emphasize the important of MyD88 in regulating numerous processes relative to skeletal muscle function and regeneration after injury.

OUR RESPONSE: We have read these two published articles. There are major differences between the experimental approach and injury models utilized in these studies and the one we are presenting here. Ischemia causes muscle injury due to a disruption of vascular supply while BaCl₂ mediated injury is achieved through direct necrosis of myofibers. Ischemia causes partial muscle injury and muscle start regenerating when new vessels are formed. Moreover, the muscle injury in response to ischemia is chronic in nature. It also causes variable injury and inflammatory response. For example, in the PLOS One manuscript, authors did not find signs of muscle injury in MyD88 KO mice after 2 week of ischemia. For our studies, we have used a well-defined acute muscle injury model which causes complete necrosis of myofibers followed by a defined pattern of inflammatory response, satellite cell activation, and myofiber regeneration. We have used both whole body MyD88-KO and myoblast-specific KO mice. We have not observed any changes in the inflammatory response and only found that myofiber CSA is reduced in KO mice. Our *in vivo* results are consistent with the cell culture studies, which demonstrate that the deletion of MyD88 drastically reduces myotube formation. While we have now cited these articles in our revised manuscript (Page # 21, ref. 60, 62), we believe that some of the differences observed between our study and these two published articles could be attributed to different experimental approaches and different models of muscle injuries utilized in each investigation.

5. The authors state they wanted to assess how MyD88 was upregulated in myoblasts but, in fact, have done little to provide insight. For example, did DM induce an increase in MyD88 mRNA levels, MyD88 transcription, or MyD88 translation or stabilization?

OUR RESPONSE: We have performed several experiments to address this issue. While we observe increased MyD88 protein levels after incubation of myoblasts in differentiation medium, there was no change in the mRNA levels of MyD88. We have now included these mRNA data as Supplementary Figure 3A. However, ongoing studies in our laboratory have provided initial evidence that a specific microRNA regulates the translation of MyD88 mRNA and myoblast fusion during myogenesis. We plan to continue working on this aspect and publish a separate manuscript in the near future.

6. Is MyD88 increased in RMS cells? This is important as the authors emphasize the RMS as a clinically relevant scenario involving the apparent activation of this process.

OUR RESPONSE: In our previous manuscript, we presented the data that MyD88 is not up-regulated in RMS cells after incubation in differentiation medium. Because RMS results do not

contribute much to the manuscript and as suggested by Reviewer # 2, we have now excluded these results from our revised manuscript.

7. Did the Myod1-Cre mice show any abnormality in skeletal muscle?

OUR RESPONSE: We did not find any abnormal phenotype in Myod1-Cre mice.

8. Figure 9 presents a critical and exciting finding that in skeletal muscle hypertrophy (independent of skeletal muscle direct injury) leads to increased MyD88 expression and a role for MyD88 in muscle hypertrophy. However, the numbers of mice (n=3) is rather small. In an era of an emphasis on rigor and reproducibility in science, this small n number raises concerns.

OUR RESPONSE: We have now repeated the experiment with 4 additional mice in each group. There are now 7 mice (n=7) in each group for this experiment (now Fig. 8B-8D).

Reviewers' comments:

Reviewer #1 (Remarks to the Author):

In their revised manuscript, Hindi et al. have reasonably answered to my initial comments and, I think, to those of the other referees.

Reviewer #2 (Remarks to the Author):

1. In general, the LOF and GOF phenotypes observed *in vivo* and *in vitro* are not visually robust compared to respective controls, especially in Figures 1D & E, 3A, 5D, 7A, 9B, and S5C. To be fair, the low resolution of images presented (the H & E stains and *in vitro* myosin images) may mask some observable differences in phenotypes. We suggest using immunostaining with an antibody that outlines the myofiber (dystrophin, laminin) for *in vivo* sections. Overall, the authors should provide images more representative of their quantitative data.

OUR RESPONSE: We agree with the reviewers that some of our images especially those stained with H&E were not of good quality. However, we would also like to mention the quality of images is considerably diminished when converting them into PDF and then inserting into the main manuscript and again making PDF of the whole manuscript. Our original images look much sharper and are of publication quality. Per reviewer's suggestion, we have now performed dystrophin or laminin staining on the muscle sections for all our *in vivo* studies. We hope reviewer will find new images of improved quality and the phenotype is now clearly visible.

Reviewer Response: The authors now show dystrophin stained images instead of the H&E images shown in the initial submission. The dystrophin images are an improvement, however I am confused as to what images were used for quantification purposes. In the Experimental Procedures it states that H&E images were used to determine CSA but in the Results section it states that 'Quantitative analysis of dystrophin-stained sections showed that....' It should be crystal clear which images were used for quantification.

a. Though the quality of the images has improved in general, the images selected do not necessarily reflect the suggested phenotypes or match their corresponding quantifications. For example in Figures 1D and 1G, there are no observable size differences in CSA between WT and MyD88f/f myofibers based on immunostaining. Though these are separate strains of mice, CSA analysis does reveal a numerical difference (~220 μm vs ~170 μm), and the authors should present appropriate images to reflect this. Similarly in Figure 6A, it is difficult to see differences in the number of eMyHC+ myofibers, which contradicts the modest findings in 6B. Maybe the authors should include arrows to indicate instances where eMyHC staining is absent within a laminin+ myofiber in MyD88-ablated muscle sections.

2. The authors state there are more mono-nucleated cells in the MyD88 KO sample in Figure 1D. They are stating this to provide evidence for lack of fusion, however this is an example of over-interpretation. First, from the images I am unable to see an increase in mono-nuclear cells. Secondly, the lineage of these mono-nuclear cells is not reported. Their interpretation would be more appropriate if they stained for myh3 (embryonic myosin) or myogenin and clearly showed these were mono-nucleated. Moreover, this observation could be due to loss of MyD88 in non-muscle cells.

OUR RESPONSE: We agree with the reviewer that our images were not of good quality and our statement about more mononucleated cells may not be correct due to lack of specific staining. We have now performed immunostaining for eMyHC, MyoD, and Pax7 on muscle sections. There were very few eMyHC+ fibers (3-4 per section) and there was no difference between WT and MyD-KO mice. We also did not find any MyoD+ cells. By contrast, we found that there was a small increase (but not statistically significant, $p=0.061$) in the number of Pax7+ satellite cells in skeletal muscle of 2-week old MyD88myoKO mice compared to corresponding control mice (Supplementary Fig. 1B and 1C).

Reviewer Response: I originally suggested an experiment to assess the lineage of the mononuclear cells to corroborate their statement about the importance of more mononuclear cells in the MyD88 KO sample. But now the authors removed this statement after assessing myh3, myod, and pax7. They see no significant difference in Pax7+ cells suggesting that their initial interpretation regarding the mononuclear cells was incorrect. The authors interpret this to mean that 'MyD88 does not affect the post-natal muscle development program'. In order to make that statement they would need to analyze an earlier time-point when expression of myod and myh3 were present in normal mice. Nonetheless, they still have not provided any evidence of reduced fusion during the neonatal period but have interpreted that this is the reason for a lower CSA.

3. Another example of over-interpretation relates to the RMS experiment. MyD88 ONLY weakly increases fusion after the cells are induced to become myogenic (through expression of a constitutively active MKK6), but essentially has no effect in cultures not myogenically activated. And the repression of MyD88 in RMS cells is not surprising given that the majority of differentiation genes are reduced (they even show a reduction in myosin). I think using this sub-optimal data to even suggest that this could be a valuable therapeutic for RMS is grossly misleading. I would even argue that this piece of data could be excluded as it does not add much to their story. Lastly, with regards to fusion, while data in the current study do support a role for MyD88 in mediating myoblast fusion in vitro, it is an overstatement to claim that MyD88 is required or essential in this process as described throughout the manuscript.

OUR RESPONSE: We agree with the reviewer that RMS cell results were just supportive and did not add much to our manuscript. Per reviewer's suggestion we have now excluded RMS data from our manuscript. As mentioned above in response to similar comments by Reviewer #1, we have now toned down our sentences. We now state that MyD88 mediates myoblast fusion instead of claiming it is essential for myoblast fusion.

4. Is there a reduction in CSA in MyD88 mKO mice after barium chloride injury? Additionally, the authors should quantify the number and percentage of regenerating myofibers to ascertain if regeneration is compromised following injury with MyD88 deletion in myoblasts. As MyD88 is essential for innate immunity, it is curious that the authors do not observe a more severe phenotype (e.g. excess infiltration of inflammatory cells, increased swelling) in their global MyD88-KO mice following injury (Fig. S5C). Could the authors explain this discrepancy? Furthermore, the authors should clarify if mice were treated with BaCl₂ at 8-12 weeks old (as stated in experimental procedures), or at 3-month old (as stated in results and figure legends). Mice at 8 weeks old may not be fully developed and data generated at this age (especially myofiber CSA) should not be included in the analyses of adult skeletal muscle regeneration.

OUR RESPONSE: We have now performed muscle injury experiment in MyD88myoKO mice and performed eMyHC staining and measured CSA. We observed a significant decrease in CSA of eMyHC+ myofibers in MyD88myoKO mice compared to controls. Please refer to new Figures 6A-C. We agree with the reviewer that MyD88 is involved in the inflammatory response in some pathological conditions. However, with the exception of reduced myofiber CSA in regenerating skeletal muscle of whole body MyD88-KO mice (now Supplementary Fig. 9), we did not find any major difference including cellular infiltrates etc. (now Supplementary Fig. 9). Since the focus of the paper is on the role of MyD88 in myoblasts, we have now specifically investigated the activation of satellite cells and inflammatory immune response in regenerating skeletal muscle of MyD88^{f/f} and MyD88myoKO mice. Our results demonstrated that the deficiency of MyD88 in myoblasts did not affect the number of satellite cells or immune cells (i.e. CD45+) in regenerating muscle of MyD88myoKO mice (Supplementary Fig. 10).

5. The authors mention that MyD88 mKO mice do not exhibit reduced CSA at 3 months of age, however they do comment further. Does this suggest that the kinetics of fusion are just slower in MyD88 mKO mice and they eventually catch up? Or do the 3-month old MyD88 mKO myofibers exhibit reduced number of myonuclei, but this number is enough to support growth? The authors could get at these questions by isolating single myofibers and quantifying nuclei/myofiber. Or

maybe this suggests that MyD88 does not directly augment fusion.

OUR RESPONSE: This is an excellent suggestion by the reviewer. We have now performed this experiment using isolated single myofibers from 3-month old MyD88f/f and MyD88myoKO mice. Our results demonstrate that the overall number of myonuclei per myofiber is indeed significantly reduced in MyD88myoKO compared to control mice yet is sufficient to support muscle growth to the level of wild type counterparts (Please refer to Supplementary Fig. 2). We have now included and discussed these results in this revised submission.

Reviewer Response: The single myofiber images presented in Suppl 2A exhibit high background and low resolution, which may mask the ability to accurately detect DAPI myonuclei. Did the authors quantitate nuclei manually or in an automated fashion using a software? The number of DAPI+ myonuclei should be normalized to some parameter (such as myofiber length). These technical limitations are quite important here given that the phenotype is relatively modest. Conceptually, while the authors conclude that the reduced number of myonuclei in their knockouts at 3 months of age is sufficient to support muscle growth, how do they reconcile differences in CSA at 2 weeks? Is fusion perturbed at 2 weeks (did they analyze this), or are there some other fusion independent mechanism(s) involved? They have not directly shown that fusion is perturbed in vivo during neonatal muscle growth.

6. Overall, the discussion is simply a rehash of the results and does not deal with the important issues. For instance, our comment #5 above was not mentioned. Additionally, why is there such a modest fusion phenotype in the MyD88-KO, especially compared to their in vitro data? Also why there is no CSA phenotype in MyD88 KO after injury (Figure 7)? Does a more pronounced muscle phenotype in MyD88-KO compared to MyD88 mKO suggest that MyD88 is also functioning in non-muscle cells to augment fusion? I realize these are difficult questions that may not indicate a direct role for MyD88 during myoblast fusion, but they should still be discussed.

OUR RESPONSE: We have now considerably improved the discussion part of our manuscript. We have also included the points suggested by the reviewer. In the last submission, we did not present data about myofiber CSA of regenerating muscle of MyD88f/f and MyD88myoKO mice. We have now performed muscle injury experiment with MyD88myoKO mice similar to what was done with MyD88-KO mice. Our experiment demonstrates that there is significant reduction in myofiber CSA in MyD88myoKO mice at 5 days post injury (now Fig. 6C) which is comparable to what we observed in MyD88-KO mice (in Supplementary Fig. 9E). We have also provided a possible explanation as to why the phenotype is modest in vivo compared to in vitro in the "Discussion" section of this revised manuscript (highlighted text on page # 26).

7. To substantiate the claim that "MyD88 promotes skeletal muscle growth in adult mice through augmenting myoblast fusion" (results), the authors need to directly assess and quantify the degree of fusion in their control and myoblast-specific KO mouse following overload.

OUR RESPONSE. This is an excellent suggestion. We have now specifically assessed the degree of fusion by labelling myoblasts with EdU in the synergistic ablation experiment. We found a significant reduction in the number of EdU+ myonuclei in overloaded muscle of MyD88myoKO mice compared to corresponding control mice. Please refer to new Figures 8E and 8F in our revised experiment.

Reviewer Response: To clearly establish reduced myoblast fusion as a mechanism for impaired muscle growth following overload, the authors should show higher magnification images of their EdU staining to clearly visualize EdU+ nuclei inside the myofiber. From the current images presented, the distinction between interstitial or myofiber EdU+ nuclei are not clearly visible. It is also very puzzling that the authors reported an average of >1 intrafiber EdU+ nuclei in control muscles after overload. They quantify the number of EdU+ nuclei per myofiber, and thus their analysis says that each myofiber has one EdU+ myonucleus although it could mean that some myofibers have multiple EdU+ nuclei but this is rare when only examining one level of the section. It has been previously reported that roughly 25-40% of myofibers contain a labeled nucleus after overload (McCarthy JJ, et al. Development 2011 and Goh Q, Millay DP eLife 2017). The EdU+ myonucle is also confusing here because it is much lower than following BaCL injury (<0.8) (Figure

6E), which is a much more robust method of inducing fusion. The authors should explain this discrepancy. Lastly, as the contribution of myoblast fusion to muscle hypertrophy is still fiercely debated, the authors should cite relevant literature to support their findings.

8. The authors clearly show that fusion, though diminished, does occur in the absence of MyD88 (Figure 2) through immunostaining and quantification of smaller myotubes with fewer nuclei. To strengthen overall findings, it would be informative to reveal the total number of myotubes for existing samples. This would provide insight on the role of MyD88 in primary fusion (fusion of myoblasts to generate nascent myotubes) compared to secondary fusion (myonuclear addition in existing myotubes) and add to understanding of MyD88 function.

OUR RESPONSE: We have now counted the number of myotubes. Please refer to Fig. 2F and 2G in our revised manuscript. We did not find any difference in the overall number of myotubes containing 2 or more nuclei between WT and MyD88-KO cultures indicating that most MyD88 deficient myoblasts are capable of undergoing primary fusion and forming nascent myotubes. However, we also observed that not all MyD88-KO myoblast underwent primary fusion and a significant number although expressed MyHC, remained mononucleated. Based on this analysis, we believe that MyD88 primarily mediates secondary fusion and is involved but to a lesser extent in mediating primary fusion. This has now been discussed in the revised manuscript (highlighted text on Page # 9 and # 22).

Reviewer Response: On page 9, the authors offer an additional interpretation that MyD88 mainly impacts secondary myoblast fusion. This should be put in context for the non-expert reader by explaining the difference between primary (myoblast – myoblast) and secondary (myoblast – myotube) myoblast fusion.

9. In Figure 5, the authors investigate the non-canonical NF- κ B pathway in myoblasts and show that in MyD88 KO cells there is a reduction of p52 cleavage. Overall the authors think that MyD88 promotes p52 cleavage. They then rescue MyD88 KO myoblasts with expression of p52, RelB, IKK α . How this rescues fusion, even in the absence of MyD88, is not clear. If the normal function for MyD88 in myoblasts is to promote cleavage of p52, then how does adding more p52 overcome the effect of loss of MyD88? Does something else cleave p52 in the absence of MyD88 but only when p52 is extremely high? The authors show an increase in p52 in 5G but do not show cleavage. The authors should deal with these issues.

OUR RESPONSE: Upon activation of non-canonical NF- κ B signaling pathway, IKK α phosphorylates p100 protein. Upon phosphorylation, p100 undergoes partial proteolytic degradation leading to the formation of p52 subunit. The p52 subunit makes a dimer with RelB through which both molecules translocate to the nucleus and bind to DNA to increase gene expression. The proteolytic processing of p100 into p52 protein is a critical step in the activation of non-canonical NF- κ B signaling. We used cDNA which express only the p52 part of the p100 subunit. In this way, the proteolytic processing step is bypassed because p52 is the only functional part of p100/p52 NF- κ B subunit. Overexpression of p52 protein is a common approach to activate non-canonical NF- κ B pathway independent of upstream signaling. Since we overexpressed p52 cDNA we showed only p52 protein levels in Fig. 4F. Overexpression of p52 does not affect the levels of p100 protein which is also evident in the uncropped gel image included in the Supplementary Fig. 11 of this manuscript.

Reviewer Response: This explanation makes sense but is missing from the text. Maybe the authors should include a similar description for the readers who are not experts on NF- κ B signaling.

Minor points:

1. Muscle weights are shown from MyD88 KO mice in Figure 1, demonstrating reduced muscle mass. However, these weights are not normalized. Additionally, it is not reported whether non-muscle tissues are also smaller? If so, it could suggest that MyD88 regulates tissue size through fusion-independent mechanisms.

OUR RESPONSE: We have now presented muscle weight normalized to whole body weight. We had

also collected spleen and kidney from these mice. As our results in Fig. 1C demonstrate, there is a significant reduction in weight in most muscle groups when normalized to total body weight, but not in kidney or spleen, suggesting that MyD88 specifically affects skeletal muscle.

2. Figure 1F was not referred to in the text.

OUR RESPONSE: This has now been referred (on Page # 6).

3. The authors should provide a reference for the use of LDH as a marker for cell death. They mention it has been extensively used on page 8 but no reference is listed.

OUR RESPONSE: We have now cited three references for this (ref. 43-45).

4. In Figure 8, though the authors used mTmG mice for Myd88 overexpression experiments, they assessed and quantified tomato+ myofibers. Hence, headings in Figures 8B and 8C should be changed to reflect that.

OUR RESPONSE: The reviewer is correct. We have now corrected this to mT+ (now Figure 7).

5. In Figure 9 legends, * was used to denote differences between corresponding shams, as well as between 14d-loaded muscle of control mice; however # should be used to depict differences for the latter finding, as shown in Figures 9C and 9D.

OUR RESPONSE: This was an oversight on our part. It has now been corrected (now Fig. 8C and 8D).

6. On page 10 of results, the authors mentioned increased mRNA levels of M-cadherin and N-cadherin, even though they are not statistically significant, but omitted Adam-12 even though that molecule demonstrated the most robust change in expression following MyD88. This should be revised accordingly.

OUR RESPONSE: We agree with the reviewer. This was again an oversight on our part. We have now mentioned this in the "Results" section (on Page # 11, 12).

7. The authors do not report the percentage of transfected cells in any of their experiments. This is an important factor for proper interpretation and could even account for some less than robust effects if they are dealing with heterogeneous cultures.

OUR RESPONSE: We used the electroporation approach for overexpression studies. We consistently get ~80% cells transfected. We have now mentioned this in the "Experimental procedure" section (Page # 31).

8. The heading "MyD88 is required for load-induced myofiber hypertrophy" is not completely accurate (Fig. 9) as ~20% increase in myofiber CSA was observed in MyD88 mKO plantaris 14 d after surgery. The heading used in the results section "MyD88 mediates overload-induced muscle hypertrophy in adult mice" is therefore more appropriate.

OUR RESPONSE: We agree with the reviewer. We have revised the heading accordingly (Page # 19).

New Minor points from reviewer:

- On page 3 the authors have included a sentence about the identification of a new fusion factor, which is appropriate. However, I don't see how the sentence fits into the paragraph especially because the preceding sentence is about control of myoblast fusion by MRFs. Then the new sentence begins with 'However'. It is an odd structure and give the reader a sense that the authors just put it at the end of paragraph without much thought. Moreover, I just saw there were two additional papers on this same gene published in Nature Communications so they should likely be cited as the authors decide whether to include this topic in their introduction.
- There is no quantification for Supplementary Fig. 4
- Why is myosin staining so much weaker in Fig. 2B but there is no difference by western blot? The myosin staining also is not ideal in Fig. 3A. Clear demarcation of myotubes is not observed.
- Perhaps the biggest issue is whether MyD88 is a direct regulator of fusion or of the steps preceding fusion. Indeed, overexpression of MyD88 increased levels of myogenin and myosin (Fig. 3E) suggesting a role in differentiation.

Reviewer #3 (Remarks to the Author):

The additional work performed as part of the revisions considerably strengthens the novel conclusions of this paper. In addition the additional controls and experiments performed to link the downstream signaling also support the novelty of the observations.

RESPONSE TO REVIEWERS' COMMENTS

We thank all the reviewers for carefully reading our manuscript and providing valuable suggestions. Thanks to the reviewers' comments, we feel our manuscript has been tremendously improved. While Reviewer # 1 and # 3 had no more comments, reviewer # 2 raised additional concerns which have been addressed in this revised version of our manuscript.

REVIEWER # 2

The reviewer's main suggestion was to analyze myoblast fusion during postnatal growth of muscle. Per your suggestion, we have also analyzed the mice at the neonatal age and have specifically studied myoblast fusion during postnatal growth. Our new results demonstrate that there is a reduction in myoblast fusion in MyD88^{myoKO} mice during post-natal growth (Supplementary Figures 2 and 3). We hope through this revision we have now addressed all the concerns of the reviewer. The changes made in the manuscript are highlighted using yellow background. Our point-wise responses to the reviewer's comments are as follows:

1. In general, the LOF and GOF phenotypes observed in vivo and in vitro are not visually robust compared to respective controls, especially in Figures 1D & E, 3A, 5D, 7A, 9B, and S5C. To be fair, the low resolution of images presented (the H & E stains and in vitro myosin images) may mask some observable differences in phenotypes. We suggest using immunostaining with an antibody that outlines the myofiber (dystrophin, laminin) for in vivo sections. Overall, the authors should provide images more representative of their quantitative data.

OUR RESPONSE: We agree with the reviewers that some of our images especially those stained with H&E were not of good quality. However, we would also like to mention the quality of images is considerably diminished when converting them into PDF and then inserting into the main manuscript and again making PDF of the whole manuscript. Our original images look much sharper and are of publication quality. Per reviewer's suggestion, we have now performed dystrophin or laminin staining on the muscle sections for all our in vivo studies. We hope reviewer will find new images of improved quality and the phenotype is now clearly visible.

Reviewer's comment: The authors now show dystrophin stained images instead of the H&E images shown in the initial submission. The dystrophin images are an improvement, however I am confused as to what images were used for quantification purposes. In the Experimental Procedures it states that H&E images were used to determine CSA but in the Results section is states that 'Quantitative analysis of dystrophin-stained sections showed that...' It should be crystal clear which images were used for quantification.

OUR RESPONSE: In this experiment, we used dystrophin-stained sections for CSA analysis. Indeed, there was hardly any difference in average myofiber CSA when we used H&E-stained or dystrophin-stained images. As the reviewer pointed out, we mentioned in the "Results" section, but we had to keep the "Experimental Procedures" section more or less the same because we used H&E-stained images for some other figures in the manuscript (i.e. Figure 8B, and Supplemental Figure 9). We have now specifically mentioned in the "Experimental Procedures" section that for quantification of CSA we used "dystrophin- or H&E stained" images. We also mentioned in the figure legends of (wherever applicable) which images were used for the quantification of myofiber CSA.

Reviewer's Comment: Though the quality of the images has improved in general, the images selected do not necessarily reflect the suggested phenotypes or match their corresponding quantifications. For example in Figures 1D and 1G, there are no observable size differences in CSA between WT and MyD88^{f/f} myofibers based on immunostaining. Though these are separate strains of mice, CSA analysis does reveal a numerical difference (~220 um vs ~170 um), and the authors should present appropriate images to reflect this.

OUR RESPONSE: We agree that the left part of Figure 1G may not be representative. We have now provided a more representative image here.

Reviewer's comment: Similarly in Figure 6A, it is difficult to see differences in the number of eMyHC⁺ myofibers, which contradicts the modest findings in 6B. Maybe the authors should include arrows to indicate instances where eMyHC staining is absent within a laminin⁺ myofiber in MyD88-ablated muscle sections.

OUR RESPONSE: Instances of laminin⁺/eMyHC⁻ were present in our previous versions of the figures. Per the reviewer's suggestion, we have added arrows for laminin⁺/eMyHC⁻ fibers in the images presented in Figure 6A.

2. The authors state there are more mono-nucleated cells in the MyD88 KO sample in Figure 1D. They are stating this to provide evidence for lack of fusion, however this is an example of over-interpretation. First, from the images I am unable to see an increase in mono-nuclear cells. Secondly, the lineage of these mono-nuclear cells is not reported. Their interpretation would be more appropriate if they stained for myh3 (embryonic myosin) or myogenin and clearly showed these were mono-nucleated. Moreover, this observation could be due to loss of MyD88 in non-muscle cells.

OUR RESPONSE: We agree with the reviewer that our images were not of good quality and our statement about more mononucleated cells may not be correct due to lack of specific staining. We have now performed immunostaining for eMyHC, MyoD, and Pax7 on muscle sections. There were very few eMyHC⁺ fibers (3-4 per section) and there was no difference between WT and MyD88-KO mice. We also did not find any MyoD⁺ cells. By contrast, we found that there was a small increase (but not statistically significant, p=0.061) in the number of Pax7⁺ satellite cells in skeletal muscle of 2-week old MyD88myoKO mice compared to corresponding control mice (Supplementary Fig. 1B and 1C).

Reviewer's comment: I originally suggested an experiment to assess the lineage of the mononuclear cells to corroborate their statement about the importance of more mononuclear cells in the MyD88 KO sample. But now the authors removed this statement after assessing myh3, myod, and pax7. They see no significant difference in Pax7⁺ cells suggesting that their initial interpretation regarding the mononuclear cells was incorrect. The authors interpret this to mean that 'MyD88 does not affect the post-natal muscle development program'. In order to make that statement they would need to analyze an earlier time-point when expression of myod and myh3 were present in normal mice. Nonetheless, they still have not provided any evidence of reduced fusion during the neonatal period but have interpreted that this is the reason for a lower CSA.

OUR RESPONSE: Our original manuscript was not a developmental biology study. Indeed we have now removed "skeletal muscle growth" from the title of the manuscript. However, per the reviewer's suggestion, we have now analyzed skeletal muscle of mice at day 5 post birth (P5). The reviewer will find that eMyHC is expressed in skeletal muscle of both control and

MyD88^{myoKO} mice at this age (Supplementary Figure 2). Importantly, we found that skeletal muscle of MyD88^{myoKO} mice contained an increased percentage of smaller-sized eMyHC⁺ cells, which contained only one nucleus in the cross-sections. We quantified the number of these mononucleated eMyHC⁺ cells and found a significant increase in 5-day old MyD88^{myoKO} mice compared to littermate control mice (Supplemental Figure 2). We have also now evaluated myoblast fusion using the EdU incorporation assay during the post-natal growth of mice. For this experiment, we gave two intraperitoneal injections of EdU in mice (at P5 and P8) and their skeletal muscle was evaluated at the age of 2 weeks. Consistent with all other results, we found a significant reduction in EdU⁺ nuclei within myofibers of MyD88^{myoKO} mice (Supplemental Figure 3). These experiments provide direct evidence that myoblast fusion is attenuated, which may be responsible for the reduced myofiber CSA in MyD88^{myoKO} mice at the age of 2 weeks.

5. The authors mention that MyD88 mKO mice do not exhibit reduced CSA at 3 months of age, however they do comment further. Does this suggest that the kinetics of fusion are just slower in MyD88 mKO mice and they eventually catch up? Or do the 3 month old MyD88 mKO myofibers exhibit reduced number of myonuclei, but this number is enough to support growth? The authors could get at these questions by isolating single myofibers and quantifying nuclei/myofiber. Or maybe this suggests that MyD88 does not directly augment fusion. **OUR RESPONSE:** This is an excellent suggestion by the reviewer. We have now performed this experiment using isolated single myofibers from 3-month old MyD88^{f/f} and MyD88^{myoKO} mice. Our results demonstrate that the overall number of myonuclei per myofiber is indeed significantly reduced in MyD88^{myoKO} compared to control mice yet is sufficient to support muscle growth to the level of wild type counterparts (Please refer to Supplementary Fig. 2). We have now included and discussed these results in this revised submission.

Reviewer's comment: The single myofiber images presented in Suppl 2A exhibit high background and low resolution, which may mask the ability to accurately detect DAPI myonuclei. Did the authors quantitate nuclei manually or in an automated fashion using a software? The number of DAPI⁺ myonuclei should be normalized to some parameter (such as myofiber length). These technical limitations are quite important here given that the phenotype is relatively modest. Conceptually, while the authors conclude that the reduced number of myonuclei in their knockouts at 3 months of age is sufficient to support muscle growth, how do they reconcile differences in CSA at 2 weeks? Is fusion perturbed at 2 weeks (did they analyze this), or are there some other fusion independent mechanism(s) involved? They have not directly shown that fusion is perturbed in vivo during neonatal muscle growth.

OUR RESPONSE: As the reviewer knows, isolated myofibers are cylindrical, long cells, each of which contains hundreds of nuclei. It is not possible for us to get a sharp image of the entire myofiber in one field using the microscope that we have. Since myonuclei are in different planes, it is also difficult to focus on all the nuclei in the same 2D image. Most investigators present a portion of myofibers for such studies. We have also now presented a portion of representative myofibers (now Supplementary Figure 4). We hope the reviewer will find our new images of better quality. In the last submission, we had provided the total number of nuclei per myofiber. Per the reviewer's suggestion, we have now provided the results in which the number of nuclei were normalized with myofiber length. We still obtained a significant reduction in the number of myonuclei in the myofibers of MyD88^{myoKO} mice compared to MyD88^{f/f} mice. This analysis was

done manually. As explained above, we have now provided direct evidence (Supplemental Figure 3) that myoblast fusion is perturbed during neonatal growth.

7. To substantiate the claim that “MyD88 promotes skeletal muscle growth in adult mice through augmenting myoblast fusion” (results), the authors need to directly assess and quantify the degree of fusion in their control and myoblast-specific KO mouse following overload.

OUR RESPONSE. This is an excellent suggestion. We have now specifically assessed the degree of fusion by labelling myoblasts with EdU in the synergistic ablation experiment. We found a significant reduction in the number of EdU+ myonuclei in overloaded muscle of MyD88myoKO mice compared to corresponding control mice. Please refer to new Figures 8E and 8F in our revised experiment.

Reviewer’s comment: To clearly establish reduced myoblast fusion as a mechanism for impaired muscle growth following overload, the authors should show higher magnification images of their EdU staining to clearly visualize EdU+ nuclei inside the myofiber. From the current images presented, the distinction between interstitial or myofiber EdU+ nuclei are not clearly visible. It is also very puzzling that the authors reported an average of >1 intrafiber EdU+ nuclei in control muscles after overload. They quantify the number of EdU+ nuclei per myofiber, and thus their analysis says that each myofiber has one EdU+ myonucleus although it could mean that some myofibers have multiple EdU+ nuclei but this is rare when only examining one level of the section. It has been previously reported that roughly 25-40% of myofibers contain a labeled nucleus after overload (McCarthy JJ, et al. Development 2011 and Goh Q, Millay DP eLife 2017). The EdU+ myonuclei is also confusing here because it is much lower than following BaCL injury (<0.8) (Figure 6E), which is a much more robust method of inducing fusion. The authors should explain this discrepancy. Lastly, as the contribution of myoblast fusion to muscle hypertrophy is still fiercely debated, the authors should cite relevant literature to support their findings.

OUR RESPONSE: We agree with the reviewer and have now provided higher magnification images. We have also put arrows that point to the EdU+ nuclei that are located inside the myofiber (Please refer to our new Figure 8E and 8F). Our previous analysis was certainly erroneous and was an oversight on our part. We have now presented the data where we have quantified the percentage of myofibers containing EdU+ nuclei. Our analysis clearly shows a significant reduction in those myofibers in the plantaris muscle of MyD88^{myoKO} mice compared with MyD88^{ff} mice after 14 days of overload-induced hypertrophy.

We also agree with the reviewer that there are a few reports that claimed that myoblast fusion does not contribute to load-induced hypertrophy. However, these reports have been disputed and more recent studies have clearly shown that myoblast fusion indeed contributes to load-induced myofiber hypertrophy at least in young adult animals. We had cited a few articles in the “Introduction” section during the previous submission of our manuscript (ref # 5-7). We have now included additional references and added a sentence about this in the “Discussion” section (Highlighted text on Page # 27 and 28).

8. The authors clearly show that fusion, though diminished, does occur in the absence of MyD88 (Figure 2) through immunostaining and quantification of smaller myotubes with fewer nuclei. To strengthen overall findings, it would be informative to reveal the total number of myotubes for existing samples. This would provide insight on the role of MyD88 in primary fusion (fusion of

myoblasts to generate nascent myotubes) compared to secondary fusion (myonuclear addition in existing myotubes) and add to understanding of MyD88 function.

OUR RESPONSE: We have now counted the number of myotubes. Please refer to Fig. 2F and 2G in our revised manuscript. We did not find any difference in the overall number of myotubes containing 2 or more nuclei between WT and MyD88-KO cultures indicating that most MyD88 deficient myoblasts are capable of undergoing primary fusion and forming nascent myotubes. However, we also observed that not all MyD88-KO myoblast underwent primary fusion and a significant number although expressed MyHC, remained mononucleated. Based on this analysis, we believe that MyD88 primarily mediates secondary fusion and is involved but to a lesser extent in mediating primary fusion. This has now been discussed in the revised manuscript (highlighted text on Page # 9 and # 22).

Reviewer's comment: On page 9, the authors offer an additional interpretation that MyD88 mainly impacts secondary myoblast fusion. This should be put in context for the non-expert reader by explaining the difference between primary (myoblast – myoblast) and secondary (myoblast – myotube) myoblast fusion.

OUR RESPONSE: In our previous version, we defined primary and secondary myoblast fusion in the “Discussion” section of the manuscript. However, we agree with the reviewer that it would be more appropriate to have this in the “Results” section. We have now moved this description to the “Results” part (Page # 10, text highlighted).

9. In Figure 5, the authors investigate the non-canonical NF-kB pathway in myoblasts and show that in MyD88 KO cells there is a reduction of p52 cleavage. Overall the authors think that MyD88 promotes p52 cleavage. They then rescue MyD88 KO myoblasts with expression of p52, RelB, IKK α . How this rescues fusion, even in the absence of MyD88, is not clear. If the normal function for MyD88 in myoblasts is to promote cleavage of p52, then how does adding more p52 overcome the effect of loss of MyD88? Does something else cleave p52 in the absence of MyD88 but only when p52 is extremely high? The authors show an increase in p52 in 5G but do not show cleavage. The authors should deal with these issues.

OUR RESPONSE: Upon activation of non-canonical NF-kB signaling pathway, IKK α phosphorylates p100 protein. Upon phosphorylation, p100 undergoes partial proteolytic degradation leading to the formation of p52 subunit. The p52 subunit makes a dimer with RelB through which both molecules translocate to the nucleus and bind to DNA to increase gene expression. The proteolytic processing of p100 into p52 protein is a critical step in the activation of non-canonical NF-kB signaling. We used cDNA which express only the p52 part of the p100 subunit. In this way, the proteolytic processing step is bypassed because p52 is the only functional part of p100/p52 NF-kB subunit. Overexpression of p52 protein is a common approach to activate non-canonical NF-kB pathway independent of upstream signaling. Since we overexpressed p52 cDNA we showed only p52 protein levels in Fig. 4F. Overexpression of p52 does not affect the levels of p100 protein, which is also evident in the uncropped gel image included in the Supplementary Fig. 11 of this manuscript.

Reviewer's comment: This explanation makes sense but is missing from the text. Maybe the authors should include a similar description for the readers who are not experts on NF-kB signaling.

OUR RESPONSE: Per the reviewer's suggestion we have now added a description about NF-kB pathways in the "Results" section (Page # 15).

New Minor points from reviewer:

Reviewer's comment: - On page 3 the authors have included a sentence about the identification of a new fusion factor, which is appropriate. However, I don't see how the sentence fits into the paragraph especially because the preceding sentence is about control of myoblast fusion by MRFs. Then the new sentence begins with 'However'. It is an odd structure and give the reader a sense that the authors just put it at the end of paragraph without much thought. Moreover, I just saw there were two additional papers on this same gene published in Nature Communications so they should likely be cited as the authors decide whether to include this topic in their introduction.

OUR RESPONSE: We have now improved the flow of this sentence. We would also like to mention that when we submitted our manuscript, none of the three articles about this new peptide (Myomixer/Myomerger/Minion) had been published. During our revision, only the first article was published by Eric Olson's group in the "Science" magazine and therefore we included that in our revised submission. The two other articles in Nature Communications appeared after our revised version of the manuscript was submitted. Therefore we could not include those articles. We have now cited all three articles in this submission.

Reviewer's comment: - There is no quantification for Supplementary Fig. 4

OUR RESPONSE: The quantification of the percentage of chimeric myotubes has now been included.

Reviewer's comment: - Why is myosin staining so much weaker in Fig. 2B but there is no difference by western blot? The myosin staining also is not ideal in Fig. 3A. Clear demarcation of myotubes is not observed.

OUR RESPONSE: Western blot is a quantitative method that provides comparative levels of a specific protein. We have done at least 3-4 independent experiments and never found any difference in the MyHC levels. We have carefully reviewed higher magnification images and do not see any difference in the MyHC staining. However, mononucleated eMyHC⁺ cells have a lower cytoplasm-to-nucleus ratio compared to myotubes. Because of this, the nuclear staining appears more dominating and overshadows the MyHC staining in myoblasts.

Reviewer's comment: - Perhaps the biggest issue is whether MyD88 is a direct regulator of fusion or of the steps preceding fusion. Indeed, overexpression of MyD88 increased levels of myogenin and myosin (Fig. 3E) suggesting a role in differentiation.

OUR RESPONSE: We have provided multiple lines of *in vitro* and *in vivo* evidence that MyD88 mediates myoblast fusion, including during the postnatal growth period. Most importantly our *in vitro* data where cultures contained pure myogenic cells clearly demonstrate that MyD88 specifically regulate fusion step and not the preceding expression of myogenic regulatory factors (Fig. 2). The knockout approach is the only one by which the physiological role of a protein can be assessed. Our results clearly show that the deletion of MyD88 blocks myoblast fusion without affecting the expression of muscle differentiation markers. In general, the overexpression approach is not physiologically relevant. We used overexpression of MyD88 to validate that increasing the amounts of MyD88 rescues fusion defects in MyD88-KO

myoblasts and augments the fusion of WT myoblasts. MyD88 is an adaptor protein that regulates the activation of a number of signaling pathways. A small increase in myogenin and MyHC upon overexpression of MyD88 could be an indirect effect of the activation of other MyD88-mediated signaling pathways under supra physiological conditions.

[Redacted]

We sincerely hope the reviewer is convinced by these highly novel findings about the role of MyD88 in myoblast fusion. Thank you again for your valuable suggestions on our manuscript.

REVIEWERS' COMMENTS:

Reviewer #2 (Remarks to the Author):

I have no further concerns regarding this work.